# Prompt-Agnostic Adversarial Perturbation for Customized Diffusion Models

**Cong Wan**
Xi'an Jiaotong University
wancong@stu.xjtu.edu.cn

**Yuhang He** *
Xi'an Jiaotong University
heyuhang@xjtu.edu.cn

**Xiang Song**
Xi'an Jiaotong University
songxiang@stu.xjtu.edu.cn

**Yihong Gong**
Xi'an Jiaotong University
ygong@mail.xjtu.edu.cn

## Abstract

Diffusion models have revolutionized customized text-to-image generation, allowing for efficient synthesis of photos from personal data with textual descriptions. However, these advancements bring forth risks including privacy breaches and unauthorized replication of artworks. Previous researches primarily center around using "prompt-specific methods" to generate adversarial examples to protect personal images, yet the effectiveness of existing methods is hindered by constrained adaptability to different prompts. In this paper, we introduce a Prompt-Agnostic Adversarial Perturbation (PAP) method for customized diffusion models. PAP first models the prompt distribution using a Laplace Approximation, and then produces prompt-agnostic perturbations by maximizing a disturbance expectation based on the modeled distribution. This approach effectively tackles the prompt-agnostic attacks, leading to improved defense stability. Extensive experiments in face privacy and artistic style protection, demonstrate the superior generalization of PAP in comparison to existing techniques. Our code will be available at https://github.com/vancyland/PAP.

## 1 Introduction

Generative methods based on diffusion models [1–4] have made significant improvements in recent years, enabling high quality text-to-image synthesis [5, 6], image editing [7], video generation [8, 9], and text-to-3D conversion [10] by *prompt engineering*. One of the most representative methods in this field is the Stable Diffusion [11, 12], which is a large-scale text-to-image model. By incorporating customized techniques such as Text Inversion [13] and DreamBooth [14], Stable Diffusion only requires fine-tuning on a few images to accurately generate highly realistic and high-quality images based on the input prompts.

Despite this promising progress, the abuse of these powerful generative methods with wicked exploitation raises wide concerns [15], especially in portrait tampering [16–18] and copyright infringement [19]. For example, in Figure 1(a), given several photos of a person, attackers can utilize diffusion models to generate fake images containing personal information, leading to reputation defamation. Even worse, attackers can easily plagiarize unauthorized artworks using diffusion models, leading to copyright and profit issues. There is an urgent technology need to protect images from diffusion model tampering.

---

*Corresponding author

38th Conference on Neural Information Processing Systems (NeurIPS 2024).

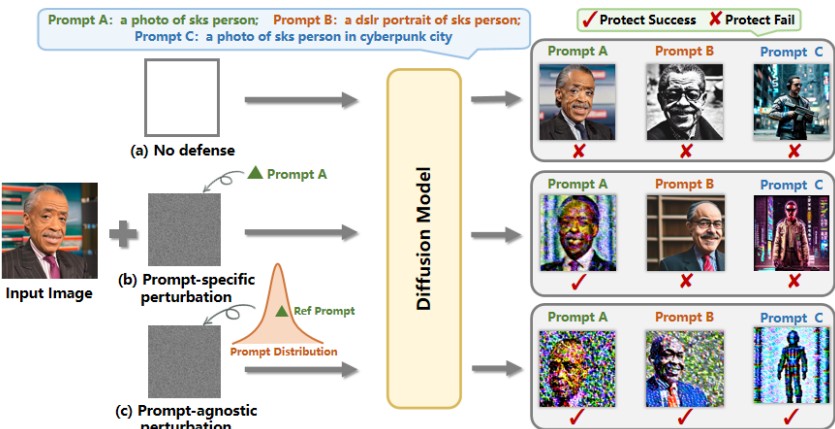

Figure 1: Illustration of a portrait with (a) no defense, (b) prompt-specific and (c) our PAP prompt-agnostic perturbation. In (a), the portrait is easily tampered with by the diffusion model. In (b), the prompt-specific methods only perform well on learned prompts (*i.e.*, Prompt A) and are fruitless to unseen prompts (*i.e.*, Prompts B and C). In (c), the proposed PAP is robust to both the seen and unseen prompts, and successfully protects the portrait from diffusion model tampering

To this end, recent research studies delve into adding human-invisible adversarial perturbations onto the images to prevent prompt-based tampering using diffusion models. The method Photoguard [20] maximizes the distance in the VAE latent space. Glaze [21] aims to hinder specific style mimicry, while Anti-DreamBooth [22]introduces an alternating training approach between the model and adversarial examples. The AdvDM series [23, 24], Adv-Diff [25] present theoretical frameworks and improved methods for attacking LDM.

These methods follow a prompt-specific paradigm, wherein they need to pre-define and enumerate the possible prompts in training, and the test prompts are required to be identical to the training ones. However, in real-world applications, once encountered with an unseen test prompt, their perturbations are inevitably futile. As shown in Figure 1(b), the perturbation trained with Prompt A fails to protect the portrait on unseen Prompts B and C at inference.

To meet this challenge, we propose a novel Prompt-Agnostic Adversarial Perturbation (PAP). Different from existing prompt-specific methods that require pre-defining and enumerating the attackers' prompts, PAP models the prompt distribution and generates prompt-agnostic perturbations by maximizing a disturbance expectation based on the prompt distribution. Specifically, using a Laplace approximation, we derive the prompts distribution in a text-image embedding feature space satisfying a Gaussian distribution, whose mean and variance can be estimated by the second-order Taylor expansion and Hessian approximation with an input reference prompt. Then, by sampling on the Gaussian prompt distribution and maximizing a disturbance expectation, we generate the PAP perturbations for the input images. As shown in Figure 1(c), the PAP perturbation is trained using a Ref Prompt, while is robust to unseen Prompts B and C at inference. To verify the efficiency of our proposed method, we conduct comprehensive experiments on three widely used benchmark datasets, including VGGFace2, Celeb-HQ and Wikiart. The experimental results show that the proposed PAP method 1) steadily and significantly outperforms existing prompt-specific perturbation on 6 widely used metrics by a large margin and 2) is robust and effective to different diffusion models, attacker prompts, and diverse datasets. These results demonstrate the efficiency and superiority of the proposed PAP method.

In summary, our contributions are as follows:

- We propose a novel Prompt-Agnostic Adversarial Perturbation (PAP) for customized text-to-image diffusion models. To the best of our knowledge, this is the first attempt at prompt-agnostic perturbation for customized diffusion models.
- We model the prompt distribution as a Gaussian distribution with Laplace approximation, where the approximation error is guaranteed. We then derive algorithms to estimate the mean and variance by Taylor expansion and Hessian approximation.

- Based on the prompt distribution modeling, we compute the prompt-agnostic perturbations by maximizing a disturbance expectation via Monte Carlo sampling.

## 2 Related Work

**Text-to-image generation models.** Text-to-image generation has evolved from early cGANs [26] and VQ-VAE [27] to advanced transformer-based models [28–33]. DALL-E [34] and GLIDE [35] demonstrate breakthrough performance in diffusion models. Recent works like Textual Inversion [13], DreamBooth [14], and ControlNet [6] further enhance generation quality through customization, user-specific fine-tuning, and additional control mechanisms.

**Gradient-based adversarial attacks.** Gradient-based adversarial attacks deceive machine learning models through perturbed inputs. FGSM [36] maximizes loss function $L(f_\theta(x + \delta), y)$ as: $\min_\theta E_{(x,y)\sim D} \max_\delta L(f_\theta(x + \delta), y)$. BIM [37] and PGD [38] enhance it through iterations. MI-FGSM [39] adds momentum constraint, NI [40] employs Nesterov acceleration, EMI [41] stabilizes with gradient averaging, and VMI [42] reduces variance via neighborhood tuning. CWA [43] addresses local optima through loss landscape analysis.

**Prompt-specific image cloaking for generation.** Early cloaking methods like Fawkes [44], Lowkey [45], and Deepfake defense [46] focus on face recognition protection. For diffusion models, Photoguard [20] maximizes VAE latent distance, Glaze [21] prevents style copying, and Anti-DreamBooth [22] employs alternating training. Recent works like UAP [47], AdvDM series [23, 24, 48], AdvTDM [49], AdvDiff [25], and Diffprotect [50] provide theoretical frameworks and improved methods for attacking diffusion models.

In contrast, our work stands apart from these approaches by focusing on the predicted prompt distribution rather than predefined prompt instances, enabling us to achieve global protection efficacy.

## 3 Prompt-Agnostic Adversarial Perturbation

### 3.1 Background and Motivation

**Prompt-based Diffusion Models.** A vanilla diffusion model [1, 3] aims to gradually transfer a simple Gaussian noise into high-quality images through a series of denoising steps. It mainly contains a forward process (*i.e.*, diffusion) and a backward process (*i.e.*, denoising). Begin with an image $x_0$, the forward process iteratively adds Gaussian noise $\epsilon \sim N(0, I)$ with a noise scheduler $1 - \alpha_t, \alpha_t \in (0, 1)_{t=1}^T$ to the input image. The $t$-step obtained noised image $x_t$ can be written as:

$$x_t = \sqrt{\bar{\alpha}_t}x_0 + \sqrt{1 - \bar{\alpha}_t}\epsilon \tag{1}$$

where $\bar{\alpha}_t = \prod_{s=1}^t \alpha_s$. When $t \to \infty$, $x_t$ is a Gaussian noise (*i.e.*, $x_t = \epsilon$). In the backward process, the objective is to learn a noise predictor $\epsilon_\theta(x_t, t)$ predicting the added Gaussian noise at each step $t$. On this basis, taking a Gaussian noise (*i.e.*, $x_t(t \to \infty)$) as input, the diffusion model denoise the image $x_t$ using $\epsilon_\theta(\cdot)$, *i.e.*, $x_{t-1} = \frac{1}{\alpha_t}(x_t - \frac{1-\alpha_t}{\sqrt{1-\bar{\alpha}_t}}\epsilon_\theta(x_t, t)) + \sigma_t \mathbf{z}, \mathbf{z} \sim N(0, I)$, and iteratively generates a high-quality image $x_0$ by $t$ denoising steps.

The prompt-based diffusion models [11] aim to generate a semantic guided image $x_0$ from the Gaussian noise $\epsilon$. To this end, in the backward process, they additionally take a text prompt $c$ as the input of noise predictor $\epsilon_\theta(x_t, t, c)$ alongside the Gaussian noise $x_t$, and align the image and text representation by cross-attention mechanism. The image generation objective of the prompt-based diffusion models can be written as:

$$\min_\theta L_{cond}(x_0, c; \theta) = E_{t,\epsilon \sim \mathcal{N}(0,1)} L(x_0, \epsilon, t, c; \theta), \tag{2}$$

where $L(x_0, \epsilon, t, c; \theta) = \|\epsilon - \epsilon_\theta(x_t, t, c)\|_2^2$.

**Prompt-Specific Perturbation.** By collecting a set of characterized images (*e.g.*, images of a certain person) and a custom-built text prompt (*e.g.* "sks"), recent approaches such as DreamBooth [14] exploit prompt-based diffusion models for customized image generation. Despite their promising progress, they raise concerns on content falsification such as portrait tampering and copyright infringement. To meet this challenge, existing diffusion-model perturbation methods [23, 22] attempt

to add an adversarial perturbation $\delta$ to the images, aiming to protect the characterized information of the images (such as a person's face) being falsified. Specifically, they often pre-define a custom text prompt $c_0$, and then optimize the adversarial perturbation $\delta$ to maximize the image generation loss function given $c_0$, which can be written as:

$$\delta^* = \arg\max_{\delta} L_{cond}(x_0 + \delta, c_0; \theta), \quad \text{s.t.} \quad |\delta|_p \leq \eta, \tag{3}$$

where $L_{cond}$ is evaluated according to Eq.(2). By adding the obtained $\delta^*$ to $x_0$, the diffusion models fail to generate high-quality images with the prompt $c_0$.

These methods, as we discussed in Section 1, are insufficient when the text prompts are different from the training prompt. As shown in Figure 1 (b), when obtaining a perturbation $\delta$ based on prompt A, the $\delta$ failed to protect the image from being modified using different prompts such as B and C. These methods compute the perturbations based on enumerated text prompt instances, *i.e.*, prompt-specific perturbation, are fruitless facing with endless attack prompts in real-world applications.

### 3.2 PAP: Prompt-Agnostic Perturbation by Prompt Distribution Modeling

Different from the existing methods that compute a prompt-specific perturbation by enumerating prompt instances, we attempt to compute a prompt-agnostic perturbation by prompt distribution modeling, wherein the obtained perturbation is robust to both seen and unseen attack prompts.

We choose the Laplace approximation as our estimator for three key reasons: 1) it typically yields a Gaussian distribution suitable for large sample sizes; 2) it simplifies computations compared to complex methods like Monte Carlo simulations, particularly when analytical forms are difficult to obtain; and 3) it aligns well with our ideal prompt embedding distribution, concentrated around extreme points.

To this end, we first model and compute a prompt distribution by Laplace approximation, wherein two estimators $\phi$ and $\psi$ are developed to compute the distribution parameters. And then we perform Monte Carlo sampling on each input distribution to maximize a disturbance expectation.

Specifically, for the prompt distribution modeling, we consider a protecting image $x_0$ as input and assume a probability-distance correlation between the attacker prompt $c$ and $x_0$, *i.e.*, the further $c$ is from $x_0$, the lower probability of $c$ is in the distribution, and vice versa. The distribution relies on $x_0$ is ambiguous, thus we introduce an auxiliary text prompt $c_0$ roughly depicting $x_0$ into the modeling. Based on this foundation, we model the prompt distribution in the embedding space as $c \in Q_{(x_0, c_0)}$, where $Q_{(x_0, c_0)}$ represents the theoretical distribution with a probability density function $p(c|x_0, c_0)$.

Based on this setup, we approximate the original distribution $Q_{(x_0, c_0)}$ using a Gaussian distribution $\hat{Q}_{(x_0, c_0)}$ by Laplace approximation, *i.e.*, $c \in \hat{Q}_{(x_0, c_0)} \sim \mathcal{N}(c_x, H^{-1})$, where $c_x = \arg\max_c p(c|x_0, c_0)$, and $H$ is the Hessian matrix of $c_x$. As we derived in Section 3.3.1, the approximation error is $\mathcal{O}(|c - c_x|^3)$, which is negligible as the sampled $c \in \hat{Q}_{(x_0, c_0)}$ is close to $c_x$.

On this basis, we propose two estimators $\phi$ and $\psi$ used to estimate $c_x$ and $H^{-1}$, respectively. For $\phi$, to compute $c_x = \arg\max_c p(c|x_0, c_0)$ that best describe $x_0$, we approximate $\hat{c}_x = \phi(x_0, \epsilon)$ by minimizing the generation loss in Eq. (2) with momentum iterations starting from $c_0$. This approach accelerates convergence and avoids getting trapped in local minima. For $\psi$, we approximate $\hat{H}^{-1} = \psi(x, \epsilon, c_0, t)$ by performing a Taylor expansion around the flattened $\hat{c}_x$ and incorporating prior information from $c_0$. In Appendixes A.2 and A.3, we have proven that the estimation errors of $c_x$ and $H^{-1}$ are with explicit upper bounds, and more detailed descriptions are provided in Section 3.3.2.

Then, we compute the prompt-agnostic adversarial perturbation $\delta$ by maximizing the expectation of $L_{cond}$ in Eq.(2) over the prompt distribution $Q_{(x_0, c_0)}$. The objective can be formulated as:

$$\delta^* = \arg\max_{\delta} E_{c \sim Q_{(x_0, c_0)}} L_{cond}(x_0 + \delta, c_0; \theta)]$$
$$= \arg\max_{\delta} \int p(c|x_0, c_0) \cdot L_{cond}(x_0 + \delta, c_0; \theta) dc, \quad \text{s.t.} \quad |\delta|_p \leq \eta, \tag{4}$$

where $p(c|x_0, c_0)$ is used to represent the probability distribution of $Q_{(x_0, c_0)}$ given the inputs

To optimize Eq.(4) from a global perspective, we devise a strategy in Section 3.4 that utilizes Monte Carlo sampling on all input distributions, including $\hat{Q}_{(x_0,c_0)}$, to maximize the disturbance expectation.

## 3.3 Modeling the Prompt Distribution $Q_{(x_0,c_0)}$

In this subsection, we first approximate the form of $Q_{(x_0,c_0)}$ and then estimate its mean and variance.

### 3.3.1 Laplace Modeling

**Definition 3.1.** *Since $x_0$ and $c_0$ are independent of c, we consider $Z = p(x_0, c_0)$ as a constant. Denote $g(c) := p(x_0, c_0|c) \cdot p(c)$, $c_x := \arg\max_c g(c)$, and $H := -\nabla\nabla_c \log g(c)|_{c_x}$ for convenience.*

We adopt Laplace approximation to model $Q_{(x_0,c_0)}$. Using Bayes' theorem, we obtain:

$$p(c|x_0, c_0) = \frac{p(x_0, c_0|c) \cdot p(c)}{p(x_0, c_0)}. \tag{5}$$

We then approximate $\log g(c)$ in Definition 3.1 around $c_x$ using a second-order Taylor expansion:

$$\log g(c) \approx \log g(c_x) - \frac{1}{2}(c - c_x)H(c - c_x)^T. \tag{6}$$

From Eq.(6) and by ignoring terms that are independent of c, we infer that

$$p(c|x_0, c_0) \propto \exp\left(-\frac{1}{2}(c - c_x)H(c - c_x)^T\right), \tag{7}$$

which means $p(c|x_0, c_0)$ could be approximated as a normal distribution, *i.e.*,

$$Q_{(x_0,c_0)}(c) \sim \mathcal{N}(c_x, H^{-1}). \tag{8}$$

The derivation is provided in Appendix A.1, wherein the error of the Gaussian approximation is the third-order derivatives of $\log p(x)$ around $c_x$, *i.e.*, $\mathcal{O}(|c - c_x|^3)$.

### 3.3.2 Parameter Estimators

**Estimator $\phi$.** According to Definition 3.1, $c_x$ is defined as the text feature that maximizes the joint probability of $x_0$ and $c_0$. As directly maximizing the likelihood is untrackable, similar to [51], we convert the likelihood maximization problem into an expectation minimization problem with a proper approximation (please kindly refer to Appendix A.2) for more details), which can be written as:

$$\hat{c}_x = \phi(x_0, \epsilon) = \arg\min_c \sum_{t=0}^{T} L(x_0, \epsilon, t, c; \theta) = \arg\min_c \sum_{t=0}^{T} \|\epsilon - \epsilon_\theta(x_t, t, c)\|_2^2 \tag{9}$$

To solve for $\hat{c}_x$ in Eq.(9) and avoid local minimal, we derive a momentum-based iterative method [39] with the initial value set as the reference prompt $c_0$:

$$m_i = \beta m_{i-1} + (1 - \beta)\nabla_c L(x, \epsilon, t, c; \theta),$$
$$c_i = c_{i-1} - r \cdot m_i, \tag{10}$$

where $m_i$ represents the momentum term at iteration $i$, and $r$, $\beta$ are learning rates.

**Estimator $\psi$.** To compute $H$ in Definition 3.1, we adopt three operations: substituting $-\log g(c)$ with the loss function $L$ in Eq.(2), incorporating prior information from $c_0$, and applying the Taylor approximation of $L$ around the flattened $\hat{c}_x$ (Detailed in Appendix A.3.1),which enable us to compute:

$$(c_0 - c_x)^T H(c_0 - c_x) = 2 \cdot (L(x, \epsilon, t, c_0; \theta) - L(x, \epsilon, t, \hat{c}_x; \theta)), \tag{11}$$

To obtain $H^{-1}$ from Eq.(11), we simplify the effective dimensionality of $H$ in Appendix A.3.2). This allows us to estimate the $H^{-1}$ using the following expression:

$$\hat{H}^{-1} = \psi(x, \epsilon, c_0, t) = \frac{\|c_0 - \hat{c}_x\|_2^2}{2 \cdot (L(x, \epsilon, t, c_0; \theta) - L(x, \epsilon, t, \hat{c}_x; \theta))} I, \tag{12}$$

where $I$ represents the identity matrix and $x$ are input images. As we analyzed in Appendix A.3.2, the cosine distance between the approximated $\hat{H}^{-1}$ and $H^{-1}$ (diagonalized assumption) is with an upper bound of 0.0909 under our standard experimental settings. This simplification significantly reduces the computational complexity with a minor approximation error.

**Algorithm 1** Prompt-Agnostic Adversarial Perturbation

---

**Input:** images $x$, reference prompt $c$, parameter $\theta$, epoch numbers $M$, $N$, learning rates $\alpha$, $r$, $\beta$, budget $\eta$, noise steps $T$, loss function $L(x, \epsilon, t, c; \theta)$ in Eq.(2)
**Output:** Adversarial examples $x_M$
Initialize $c_0 = c$, $x_0 = x$, $m_0 = 0$, $\epsilon_c \sim N(0, I)$
**for** $j = 0$ to $N - 1$ **do**
    Sample $t_c \in U(0, T)$,
    Compute gradient $g_c = \nabla_{c_j} L(x_i, \epsilon_c, t_c, c_j; \theta)$
    Compute momentum $m_{j+1} = \beta m_j + (1 - \beta) g_c$
    Update $c_{j+1} = c_j - r \cdot m_j$
**end for**
**for** $i = 0$ to $M - 1$ **do**
    Sample $\epsilon \sim N(0, I)$, $t \in U(0, T)$
    Sample $c \sim N(c_N, \frac{||c_0 - c_x||_2^2}{2 \cdot (L(x_i, \epsilon, t, c_0; \theta) - L(x_i, \epsilon, t, c_N; \theta))} I)$
    Compute gradient $g_x = \nabla_{x_i} L(x_i, \epsilon, t, c; \theta))$
    Update $x_{i+1} = \text{clip}_{x_0, \eta}(x_i + \alpha \cdot \text{sgn}(g_x))$
**end for**

---

### 3.4 Maximizing the disturbance expectation

In this subsection, we devise the strategy of maximizing the disturbance expectation based on the modeled prompt distribution, thereby obtaining the final PAP algorithm outlined in Algorithm 1.

**Sampling Distributions for maximization.** To maximize the optimization objective Eq. (4) from a global perspective, we adopt Monte Carlo sampling on all input distributions, including $Q_{(x_0, c_0)}$. Drawing inspiration from established adversarial attack methods [23, 36, 52], we iteratively sample values for $t$, $\epsilon$, and $\epsilon_c$. Subsequently, we perform a gradient ascent step of $L(x, \epsilon, t, c; \theta)$ with respect to $x$, which can be summarized as:

$$x_{i+1} = x_i + \alpha \cdot \text{sgn}(\nabla_{x_i} L(x_i, \epsilon, t, c; \theta) \big| \epsilon \sim N(0, I), t \in U(0, T), c \sim Q_{(x_0, c_0)}), \tag{13}$$

where $\text{sgn}(\cdot)$ refers to the sign function, and $\alpha$ controls the step size of the gradient ascent.

**Further Discussion.** To further enhance the effectiveness of our proposed algorithm, we seamlessly integrate it with other techniques in Appendix C, such as ASPL [22], to reach a better performance. We also discuss modifying the perturbation space with the tanh function for smoother optimization with better flexibility [52] than clipping-based constraints in Appendix B. Lastly, Appendix H explores the limitations and future directions of our model.

## 4 Experiments

In this section, we experimentally compare PAP with other protection methods for customized models, specifically targeting DreamBooth, the leading customized text-to-image diffusion model for personalized image synthesis. DreamBooth customizes models by reconstructing images using a generic prompt that includes pseudo-words like "sks," while also addressing overfitting and shifting through a prior preservation loss. We evaluate PAP on various tasks, including privacy and style protection, using diverse datasets.

### 4.1 Experimental setup

**Datasets**. Our experiments involve face generation tasks using CelebA-HQ [53] and VGGFace2 [54] datasets, as well as style imitation task using the Wikiart dataset [55]. For CelebA-HQ and VGGFace2, we select subsets of 600 images respectively, with each of 12 photos from an identical individual. For Wikiart, we choose 100 paintings, with each set consisting of 20 paintings from the same artist.

**Implementation Details**. We optimize adversarial perturbations over 50 training steps and 20 text sampling steps. The step size of image, text and momentum is set to 1/255 and 0.001, 0.5 respectively, and the default noise budget is 0.05. For the Dreambooth, LoRA ,TI models, we train the models

with a learning rate of $5 \times 10^{-7}$ and batch size 4. We perform 1,000 iterations of fine-tuning with SD1.5 as a pretrained model. The PAP method takes approximately 4 to 6 minutes to complete on an NVIDIA A800 GPU with 80GB memory. To assess defense robustness, we simulate diverse attacker prompts using ten inference sentences in Table 7 that cover a wide range of possibilities. All experimental results presented below are based on the average metrics obtained from ten sentences at default. This approach provides a comprehensive evaluation of overall protection capability with smoother metric variations.

**Evaluation Metrics**. We utilize six metrics, categorized into three aspects. For a more detailed introduction, please refer to Appendix E.4.

1) **Image-to-Image Similarity**: We adopt the following metrics to assess image similarity: CLIP Image-to-Image Similarity (CLIP-I) [56], Fréchet Inception Distance (FID), and Learned Perceptual Image Patch Similarity (LPIPS) [57]. Lower CLIP-I ($\downarrow$) and LPIPS ($\downarrow$), and higher FID ($\uparrow$) indicate better defense performance;

2) **Text-to-Image Similarity**: CLIP measures the coherence between the test prompt and generated images, with lower scores ($\downarrow$) indicating worse alignment;

3) **Image Quality**: BRISQUE [58] evaluates image quality using statistical features, with higher scores ($\uparrow$) indicating worse quality. LAION aesthetic predictor [59] assesses the aesthetic quality of images based on visual features, with lower scores ($\downarrow$) indicating worse aesthetics.

Table 1: Comparison with other adversarial perturbation methods on the face generation task (including Celeb-HQ and VGGFace2 datasets, training prompt "a photo of sks person") and style imitation task (including Wikiart dataset, training prompt "a sks painting") using ten different test prompts, where the reported metric values are the average across these ten test prompts.

| Dataset | Method | FID ($\uparrow$) | CLIP-I ($\downarrow$) | LPIPS ($\uparrow$) | LAION ($\downarrow$) | BRISQUE ($\uparrow$) | CLIP ($\downarrow$) |
|---|---|---|---|---|---|---|---|
| | Clean | 124.2 | 0.7844 | 0.4776 | 6.082 | 28.12 | 0.3368 |
| | AdvDM | 217.1 | 0.6728 | 0.5682 | 5.721 | 34.19 | 0.2905 |
| Celeb-HQ | Anti-DB | 233.3 | 0.6371 | 0.5924 | 5.497 | 35.89 | 0.2800 |
| | IAdvDM | 159.1 | 0.6955 | 0.5303 | 5.768 | 31.77 | 0.2699 |
| | PAP(Ours) | **249.9** | **0.5539** | **0.6730** | **5.280** | **36.95** | **0.2543** |
| | Clean | 230.3 | 0.6567 | 0.5471 | 5.889 | 25.67 | 0.3254 |
| | AdvDM | 243.8 | 0.5931 | 0.6775 | 5.551 | 31.41 | 0.2823 |
| VGGFace2 | Anti-DB | 273.0 | 0.5483 | 0.6960 | 5.334 | 28.95 | 0.2766 |
| | IAdvDM | 248.6 | 0.5802 | 0.6798 | 5.597 | 32.93 | 0.2835 |
| | PAP(Ours) | **288.5** | **0.5164** | **0.7023** | **5.127** | **35.02** | **0.2577** |
| | Clean | 198.7 | 0.7715 | 0.6193 | 6.367 | 27.34 | 0.3515 |
| | AdvDM | 392.7 | 0.6498 | 0.7606 | 5.949 | 33.54 | 0.3026 |
| Wikiart | Anti-DB | 386.4 | 0.6462 | 0.7396 | 5.715 | 31.01 | 0.2997 |
| | IAdvDM | 390.0 | 0.6550 | 0.7149 | 5.996 | 35.30 | 0.3037 |
| | PAP(Ours) | **448.3** | **0.5641** | **0.7782** | **5.490** | **38.47** | **0.2654** |

## 4.2 Comparison with State-of-the-Art Methods

### 4.2.1 Face Privacy Protection

We first conduct experiments in preserving face privacy. During training, the reference and training prompt are both set as "a photo of sks person". Then we use ten prompts related to sks person to evaluate the models' ability to synthesize images. The average results are presented in Table 1, showcasing the superior performance of our method in terms of all metrics on both the CelebA-HQ and VGGFace2 datasets. For example, on the Celeb-HQ dataset, our method exceeds the second-best Anti-DB by **13.55%** in LPIPS, and reduces by **13.06%** and **3.948%** in CLIP-I and LAION. Also, on the VGGFace2 dataset, our method exceeds the second-best method by **6.347%** in BRISQUE, and reduces by **5.818%** and **6.833%** in CLIP-I and CLIP. These results highlight the effectiveness of our method in preserving face privacy as well as robustness to datasets and various prompts' attacks. In Figure 2 (left), we visualize some of the comparative protection results.

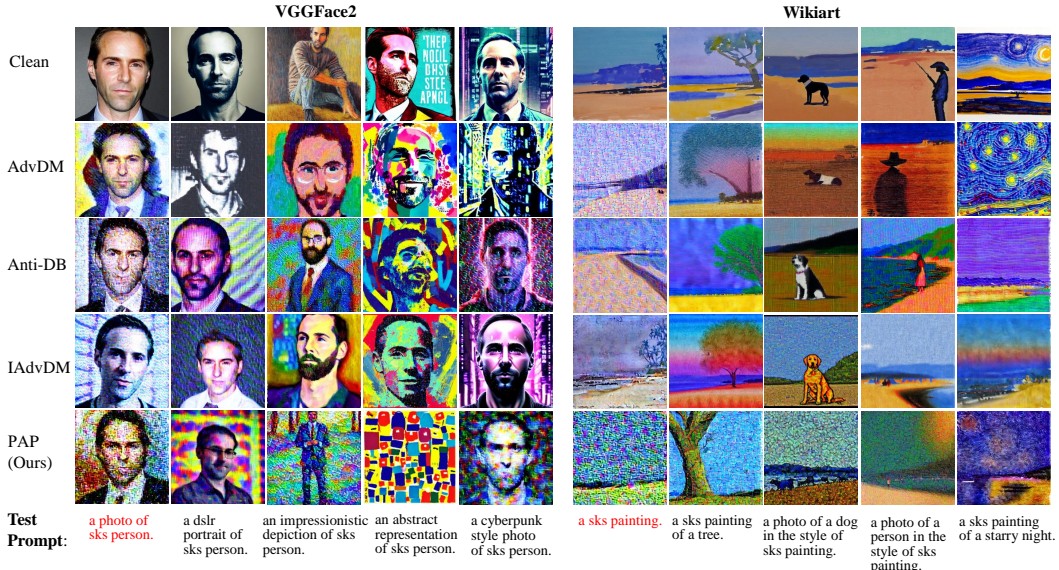

Figure 2: Qualitative defense results of different methods in VGGFace2 (left) and Wikiart (right). Each row represents a method, and each column represents a different test prompt (shown at the bottom). The adversarial examples generated by our method effectively defend against all prompts in both datasets. In contrast, other baselines primarily focus on protecting the fixed prompt (the first column), resulting in compromised defense for other prompts.

Table 2: Ablation study of text sampling steps.

| Text sampling steps | FID ($\uparrow$) | CLIP-I ($\downarrow$) | LPIPS ($\uparrow$) | LAION ($\downarrow$) | BRISQUE ($\uparrow$) | CLIP ($\downarrow$) | Cycle Time |
|---|---|---|---|---|---|---|---|
| 0  | 386.4 | 0.6462 | 0.7396 | 5.715 | 31.01 | 0.2997 | 0.3s |
| 10 | 430.8 | 0.5873 | 0.7665 | 5.577 | 36.24 | 0.2710 | 5.0s |
| 15 | 448.3 | 0.5641 | 0.7782 | 5.490 | **38.47** | 0.2654 | 7.4s |
| 20 | 457.5 | 0.5562 | 0.7854 | **5.466** | 38.33 | 0.2591 | 9.6s |
| 25 | **462.8** | **0.5481** | **0.7901** | 5.508 | 38.37 | **0.2552** | 11.9s |

### 4.2.2 Style Imitation

We also evaluate methods' ability to prevent artistic style imitation using the Wikiart dataset. The reference and training prompt are both set as "a sks painting". Ten prompts related to the style of "sks" painting are used to evaluate the model's performance and robustness. It is observed in Table 1 that our method achieves a higher FID, LPIPS, BRISQUE and lower CLIP-I, LAION and CLIP compared to existing methods. For instance, the FID and BRISQUE of our method outperforms that of others by at least **14.16%** and **8.980%** while the CLIP-I and LAION reduce by at least **12.71%** and **3.94%** These results demonstrate the effectiveness of our method in preventing style imitation as well as robustness to attacks with different prompts. In Figure 2 (right), we visualize some of the comparative protection results for Wikiart dataset. More visualized results are demonstrated in Appendix I.

### 4.3 Ablation Study

**Text Sampling Steps.** We evaluate PAP under different text sampling steps $N$ (ranging from 0 to 25) used for sampling the prompt $c$ during training. We use the Wikiart dataset and "a sks painting" prompt to conduct this ablation experiment (see Table 2). Taking into account the cycle time and model performance, we set the text sampling step $N$ to 15.

**Inference Prompt Combination.** To analyze the effect of prompt variation, we design combinations of prompt categories and quantity: $4\times20$, $8\times10$, $10\times8$, $16\times5$, $20\times4$. This keeps the total number of generated images constant while varying the number of prompt categories, allowing us to isolate the effect of prompt categories on the results. As Figure 3 shows, PAP consistently surpasses all other comparison methods and maintains a robust defense against changes in prompt categories.

**Sensitivity to the Initial Prompt.** In Table 3, we present the outcomes when using the initial prompt "", showcasing a significant decline in performance compared to the original PAP. This decline is attributed to: a) the initial prompt serves as a crucial prior for estimating parameters, facilitating rapid iteration (only 20 steps) to achieve a reliable approximation; b) it is involved in the modeling of $H$ estimation, where the approximate expression for $H$ is based on the Taylor expansion modeling of the relevant parameters.

Table 3: Results with the initial prompt ""

| Dataset | FID ($\downarrow$) | CLIP-I ($\uparrow$) | LPIPS ($\downarrow$) | LAION ($\downarrow$) | BRISQUE ($\uparrow$) | CLIP ($\downarrow$) |
|---------|------|--------|-------|-------|---------|------|
| Celeb-HQ | 154.57 | 0.72 | 0.50 | 5.83 | 30.18 | 0.32 |
| VGGFace2 | 232.34 | 0.61 | 0.60 | 5.65 | 29.20 | 0.29 |
| Wikiart | 320.79 | 0.69 | 0.71 | 5.72 | 31.88 | 0.30 |

**Pseudo-word.** In our experiments, we conduct evaluations using the commonly used pseudo-word "sks," which is representative but may not cover all possible cases. To further validate our method, we included additional less commonly used pseudo-words. Results in Table 12 clearly demonstrates that our method consistently outperforms the others with other pseudo-word.

**Evaluate the approximating $H$.** We conduct a simple experiment by directly adding Gaussian noise (with variances 1, 5, 10, and 20) to the input to evaluate the value of approximating $H$. As shown in Table 4, our proposed PAP method, using the variance estimate $\sigma^2$, achieves the best performance across all metrics. Specifically, it outperforms the second-best method by 3% (LPIPS), 2% (FDFR), 3% (ISM), and 2.27% (BRISQUE). These findings underscore the necessity of estimating variance $\sigma^2$ to generate more effective adversarial perturbations.

Table 4: Simple Baseline of Adding Gaussian Noise to the Input on VGGFace2 Dataset

| Variance | LPIPS ($\downarrow$) | FDFR ($\downarrow$) | ISM ($\uparrow$) | BRISQUE ($\uparrow$) |
|---------|-------|-------|------|---------|
| 1 | 0.67 | 0.64 | 0.38 | 31.92 |
| 5 | 0.67 | 0.65 | 0.38 | 32.75 |
| 10 | 0.66 | 0.62 | 0.40 | 29.21 |
| 20 | 0.64 | 0.60 | 0.44 | 27.01 |
| H | 0.70 | 0.67 | 0.34 | 35.02 |

**Other Customized Models.** To verify the robustness of proposed methods to fine-tuning methods, we apply PAP to Textual Inversion and DreamBooth with LoRA. LoRA [60], a widely used efficient low-rank personalization method, poses concerns due to its strong few-shot capability that enables unauthorized artistic style copying. Textual Inversion [13] learns customized concepts by simply optimizing a word vector instead of finetuning the full model. Table 5 demonstrates that PAP effectively defends against both methods, highlighting our efficacy in countering various personalization techniques.

**Noise Budget.** We conduct experiments to explore the impact of noise budget $\eta$ on PAP's defense performance (see Table 11 in Appendix F.4). A noise budget of 0.05 is effective and adopted.

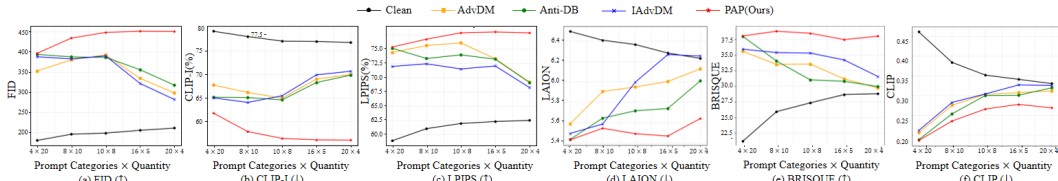

Figure 3: Defense performance of different methods in prompt variation settings. The x-axis represents the number of prompt categories multiplied by the number of generated images per prompt: 4×20, 8×10, 10×8, 16×5, and 20×4. The y-axis displays the values of different metrics.

Table 5: Robustness of our method to different fine-tuning models.

|  | Def.? | FID ($\uparrow$) | CLIP-I ($\downarrow$) | LPIPS ($\uparrow$) | LAION ($\downarrow$) | BRISQUE ($\uparrow$) | CLIP ($\downarrow$) |
|---|---|---|---|---|---|---|---|
| LoRA | $\times$ | 152.5 | 0.7944 | 0.5929 | 6.227 | 18.25 | 0.4379 |
| LoRA | $\checkmark$ | **375.8** | **0.6271** | **0.7673** | **5.534** | **41.16** | **0.2440** |
| TI | $\times$ | 144.3 | 0.8007 | 0.5882 | 6.391 | 13.21 | 0.4533 |
| TI | $\checkmark$ | **281.8** | **0.6605** | **0.7220** | **5.588** | **38.94** | **0.2462** |

## 4.4 Extending Experiments

**DiffPure.** DiffPure [61] utilizes SDEdit [62] to purify adversarial images by adding noise and denoising them using diffusion models. In Table 6, we conduct experiments on the Wikiart dataset using DiffPure (t=100). We can see that, 1) compared to *No Defense*, *PAP+DiffPure* achieves much better adversarial perturbation performance (0.565($\downarrow$), 3.38($\uparrow$), 0.0408($\downarrow$) advances on LAION, BROSQUE and CLIP metrics). 2) Compared to other methods+*DiffPure*, *PAP+DiffPure* still achieves the best performance on all metrics.

Table 6: Performance comparison after applying DiffPure with changes in ().

|  | FID ($\uparrow$) | CLIP-I ($\downarrow$) | LPIPS ($\uparrow$) | LAION ($\downarrow$) | BRISQUE ($\uparrow$) | CLIP ($\downarrow$) |
|---|---|---|---|---|---|---|
| AdvDM+DiffPure | 301.22(91.48-) | 0.71(0.06+) | 0.70(**0.06-**) | 6.217(0.268+) | 28.58(4.96-) | 0.3376(0.0350+) |
| Anti-DB+DiffPure | 335.94(50.46-) | 0.69(0.04+) | 0.68(**0.06-**) | 5.991(0.276+) | 30.12(0.89-) | 0.3410(0.0413+) |
| IAdvDM+DiffPure | 271.02(**118.98-**) | 0.72(0.01+) | 0.68(0.03-) | 6.227(0.231+) | 28.00(7.30-) | 0.3489(0.0452+) |
| PAP+DiffPure | **379.60**(68.70-) | **0.64(0.08+)** | **0.72(0.06-)** | **5.802(0.312+)** | **30.72(7.75-)** | **0.3107(0.0453+)** |
| No Defense | 198.71 | 0.77 | 0.62 | 6.367 | 27.34 | 0.3515 |

**Preprocessing.** A recent study [63] reveals that current data protections in text-to-image models are fragile and demonstrate limited robustness against data transformations like JPEG compression. To assess the resilience of our proposed Prompt-Agnostic Adversarial Perturbation (PAP) method, we conduct targeted evaluations using the LAION and BRISQUE metrics. Despite a slight decrease in performance, our method still achieves favorable outcomes in terms of image quality metrics For a detailed discussion and results, please see Appendix F.6.

## 5 Conclusion

This work mitigates risks from misusing customized text-to-image diffusion models. We introduce subtle perturbations optimized from a modeled prompt distribution, fooling such models for any prompt. Demonstrating resilience against diverse attacks, our framework surpasses prior prompt-specific defenses through robustness gains. By efficiently perturbing content via a distribution-aware method, our contributions effectively safeguard images from diffusion model tampering under unknown prompts.

## Acknowledgements

This work was funded by the NSFC under Grant No.U21B2048 and No.62302382 and Shenzhen Key Technical Projects under Grant CJGJZD2022051714160501, China Postdoctoral Science Foundation No.2024M752584, the Fundamental Research Funds for the Central Universities No.xzy012024067.

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

# A Derivation of Prompt Distribution Modeling

In this section, our goal is to develop an algorithm for modeling the prompt distribution given a reference prompt $c_0$ and protecting images $x_0$. Since the exact form of this distribution is impractical to derive, we employ the idea of Laplace approximation. By considering a second-order Taylor expansion at a critical point, we approximate the distribution near this critical point as a Gaussian distribution. To reduce computational complexity, we make assumptions and approximate estimates for the mean and variance. The algorithm is presented in Algorithm 2.

---

**Algorithm 2** Prompt Distribution Modeling

---

**Input:** images $x$, reference prompt $c$, parameter $\theta$, epoch numbers $N$, learning rates $r$, $\beta$, noise $\epsilon$, loss function $L(x, \epsilon, t, c; \theta)$ in Eq.(2).
**Output:** Mean $c_N$, square $H^{-1}$
Initialize: $c_0 = c$, $x_0 = x$, $m_0 = 0$
**for** $j = 0$ **to** $N - 1$ **do**
    Sample $t$
    Compute gradient $g_c = \nabla_{c_j} L(x, \epsilon, t, c_j; \theta)$
    Compute momentum $m_{j+1} = \beta m_j + (1 - \beta) g_c$
    Update $c_{j+1} = c_j - r \cdot g$
**end for**
Compute $\hat{H}^{-1} = \frac{||c_0 - c_N||_2^2}{2 \cdot (L(x, \epsilon, t, c_0; \theta) - L(x, \epsilon, t, c_N; \theta))}$

---

In the following, we will provide a mathematical justification for the algorithm's validity.

## A.1 Laplace Approximation Details

### A.1.1 Motivation and Basic Idea

Laplace approximation is a simple yet widely used method for approximating probability distributions, especially in the context of Bayesian inference and machine learning. The main idea behind Laplace approximation is to approximate a target distribution $p(x)$ by a Gaussian distribution $\mathcal{N}(\mu, \Sigma)$, where the mean $\mu$ and covariance $\Sigma$ are determined by matching the extreme point and curvature of the target distribution at its extreme point.

Specifically, let $c_x$ be the maximum point of $p(x)$, i.e., $c_x = \arg\max_x p(x)$. Laplace approximation uses a second-order Taylor expansion of $\log p(x)$ around $c_x$ to construct the approximate Gaussian distribution:

$$\log p(x) \approx \log p(c_x) - \frac{1}{2}(x - c_x)^T H(c_x)(x - c_x) + \mathcal{O}(|x - c_x *|^3), \tag{14}$$

where $H(c_x)$ is the Hessian matrix (the matrix of second-order partial derivatives) of $\log p(x)$ evaluated at $c_x$. Exponentiating both sides and ignoring the higher-order terms, we obtain the Laplace approximation:

$$p(x) \approx \mathcal{N}(c_x, -H(c_x)^{-1}). \tag{15}$$

### A.1.2 Error Analysis

The error of the Laplace approximation comes from neglecting the higher-order terms in the Taylor expansion. The approximation error can be quantified by the remainder term $\mathcal{O}(|x - c_x|^3)$, which represents the third and higher-order derivatives of $\log p(x)$ evaluated at some point between $x$ and $c_x$.

More precisely, let $\xi(x)$ denote the remainder term, then the Laplace approximation can be written as:

$$p(x) = \mathcal{N}(c_x, -H(c_x)^{-1}) \exp(\xi(x)), \tag{16}$$

where $\xi(x)$ satisfies $\lim_{|x-c_x|\to 0} \xi(x)/|x-c_x|^3 = 0$. This means that the relative error of the Laplace approximation is $\mathcal{O}(|x-c_x|^3)$ in a neighborhood of $c_x$.

In general, the accuracy of the Laplace approximation depends on the smoothness and behavior of the target distribution $p(x)$ around its mode $c_x$. If $p(x)$ is sufficiently smooth and concentrated around $c_x$, the higher-order terms in the Taylor expansion become negligible, and the Laplace approximation provides a good approximation. However, if $p(x)$ has heavy tails or is multi-modal, the approximation error can be significant, especially in regions far away from $c_x$.

### A.2 Estimation of $c_x$

From Definition 3.1 and derivations in AdvDM [23], we have:

$$
\begin{aligned}
\log g(c) &= \log p(c) + \log p(x_0, c_0|c) \\
&= \log p(c) + \mathbb{E}_t[\log p(x_{t-1}|c, x_t)],
\end{aligned}
\tag{17}
$$

where $x_t$ is defined in Eq.(1).

Since minimizing the diffusion loss to maximize likelihood is a common practice, we have:

$$
\begin{aligned}
c_x &= \arg\max_c \mathbb{E}_t[\log p(x_{t-1} \mid c, x_t)] \\
&= \arg\min_c \mathbb{E}_{t,\epsilon\sim\mathcal{N}(0,1)} L_{cond}(x_0 + \delta, c; \theta).
\end{aligned}
\tag{18}
$$

To reduce computational cost, we approximate Eq.(18) with bounded error, and estimate $c_x$ using a single sample of $\epsilon$ instead of averaging. The detailed derivation and justification can be found in Appendix A.2.1 and A.2.2. The final estimation of $c_x$ can be derived as follows:

$$
\hat{c}_x = \arg\min_c \sum_{t=0}^{T} L(x, \epsilon, t, c; \theta).
\tag{19}
$$

Empirical ablation experiments validate the effectiveness of this estimation, as demonstrated in Table 10.

To solve for $\hat{c}_x$ and avoid local optima, we employ an iterative method with momentum to estimate $c_x$ instead of directly solving for a local minimum. In this approach, we leverage the ensemble of loss functions from different time steps in Eq. (2). Inspired by transferable adversarial attacks [39], we aim to find a transferable $c_x$ that navigates the diverse loss landscape using the momentum iteration algorithm.

$$
\begin{aligned}
m_i &= \beta m_{i-1} + (1-\beta)\nabla_c L(x, \epsilon, t, c; \theta), \\
c_i &= c_{i-1} - r \cdot m_i.
\end{aligned}
\tag{20}
$$

Here, $m_i$ represents the momentum term at iteration $i$. The details of the information about the transferable adversarial attacks can be found in Appendix D.

Additionally, considering that $c_0$ serves as a reference prompt input that is typically expected to be highly correlated with the content of the image, we assume that $c_0$ and $c_x$ are very close in the textual space. Then the iterative solution for $c_x$ can be initialized from $c_0$.

### A.2.1 Upper bound of $c_x$ estimation error.

**Theorem A.1.** *Assume $g : \mathbb{R}^{m\times m} \to \mathbb{R}$ is Lipschitz continuous under $L_1$ norm. Then, as $n \to \infty$, we have*

$$
(\frac{1}{n}\sum_{i=1}^{n} g(x_i) - g\left(\frac{\sqrt{n}}{n}\sum_{i=1}^{n} x_i\right)) < 2L \cdot \sqrt{\frac{2}{\pi}}
\tag{21}
$$

*where $x_i \overset{i.i.d.}{\sim} \mathcal{N}(0, I)$ for $i = 1 : n$.*

*Proof.* The Lipschitz continuity condition for the 1-norm can be expressed as follows: For any $x, y \in \mathbb{R}^{m\times m}$, there exists a constant $L > 0$ such that $\|g(x) - g(y)\|| \le L\|x - y\|$. By taking the

difference of the two equations in Eq.(21), we have:

$$
\begin{aligned}
\lim_{n\to\infty} & \left\| \frac{1}{n} \sum_{i=1}^{n} g(x_i) - g\left( \frac{\sqrt{n}}{n} \sum_{i=1}^{n} x_i \right) \right\| \\
= \lim_{n\to\infty} & \left\| \frac{1}{n} \sum_{i=1}^{n} \left( g(x_i) - g\left( \frac{\sqrt{n}}{n} \sum_{i=1}^{n} x_i \right) \right) \right\| \\
\leq \lim_{n\to\infty} & \frac{1}{n} \sum_{i=1}^{n} \left\| g(x_i) - g\left( \frac{\sqrt{n}}{n} \sum_{i=1}^{n} x_i \right) \right\| \\
\leq \lim_{n\to\infty} & \frac{L}{n} \sum_{i=1}^{n} \left\| x_i - \frac{\sqrt{n}}{n} \sum_{i=1}^{n} x_i \right\| \\
= \lim_{n\to\infty} & \frac{L}{n} \left( 2 - \frac{2}{\sqrt{n}} \right) \sum_{i=1}^{n} \left\| \frac{x_i - \frac{\sqrt{n}}{n} \sum_{i=1}^{n} x_i}{\sqrt{2 - \frac{2}{\sqrt{n}}}} \right\| .
\end{aligned}
\tag{22}
$$

Notice that for any $i$ from 1 to $n$,

$$
\left( \frac{x_i - \frac{\sqrt{n}}{n} \sum_{i=1}^{n} x_i}{\sqrt{2 - \frac{2}{\sqrt{n}}}} \right) \sim N(0, I).
\tag{23}
$$

So we have:

$$
\left| \frac{x_i - \frac{\sqrt{n}}{n} \sum_{i=1}^{n} x_i}{\sqrt{2 - \frac{2}{\sqrt{n}}}} \right| \sim F(0, I).
\tag{24}
$$

The folded normal distribution, denoted by $F$, has a probability density function (PDF) given by

$$
f_Y(x; \mu, \sigma^2) = \frac{1}{\sqrt{2\pi\sigma^2}} \left( e^{-\frac{(x-\mu)^2}{2\sigma^2}} + e^{-\frac{(x+\mu)^2}{2\sigma^2}} \right),
\tag{25}
$$

for $x \geq 0$, and 0 elsewhere. The PDF can be simplified as

$$
f(x) = \sqrt{\frac{2}{\pi\sigma^2}} e^{-\frac{(x^2+\mu^2)}{2\sigma^2}} \cosh\left( \frac{\mu x}{\sigma^2} \right),
\tag{26}
$$

where $\cosh$ is the hyperbolic cosine function. The cumulative distribution function (CDF) is given by

$$
F_Y(x; \mu, \sigma^2) = \frac{1}{2} \left[ \mathrm{erf}\left( \frac{x+\mu}{\sqrt{2\sigma^2}} \right) + \mathrm{erf}\left( \frac{x-\mu}{\sqrt{2\sigma^2}} \right) \right],
\tag{27}
$$

for $x \geq 0$, where $\mathrm{erf}()$ is the error function. The expression simplifies to the CDF of the half-normal distribution when $\mu = 0$.

The mean of the folded distribution is given by

$$
\mu_Y = \sigma \sqrt{\frac{2}{\pi}} \left( 1 - e^{-\frac{\mu^2}{2\sigma^2}} \right) + \mu \left[ 1 - 2\Phi\left( -\frac{\mu}{\sigma} \right) \right],
\tag{28}
$$

where $\Phi(x)$ is the standard normal cumulative distribution function.

The variance can be easily expressed in terms of the mean:

$$
\sigma_Y^2 = \mu^2 + \sigma^2 - \mu_Y^2
\tag{29}
$$

Denote the final expression in Eq. (22) as $Y$, we can obtain the following results as $n$ approaches infinity:

$$E(Y) = \lim_{n \to \infty} \frac{L}{n} \left( 2 - \frac{2}{\sqrt{n}} \right) \cdot (n-1) \cdot \sqrt{\frac{2}{\pi}} = 2L \cdot \sqrt{\frac{2}{\pi}},$$

$$\text{Var}(Y) = \lim_{n \to \infty} \left( \frac{L}{n} \left( 2 - \frac{2}{\sqrt{n}} \right) \right)^2 \cdot (n-1) \cdot \left( 1 - \frac{2}{\pi} \right) = 0.$$

(30)

This means that the upper bound for our estimation error is $2L \cdot \sqrt{\frac{2}{\pi}}$. $\qquad\square$

### A.2.2 Single sample for $\epsilon$

Based on the theorem's conclusion, we have the flexibility to choose between weighting before the forward pass or weighting after the forward pass during Gaussian sampling, while ensuring that their errors are within a certain upper bound. By introducing the scaling factor $\sqrt{n}$, we are able to preserve the distribution characteristics of the input samples, which is crucial for subsequent diffusion processes.

An easy corollary of Theorem A.1 further states:

**Corollary A.2.** *Let $L(\epsilon)$ denotes Eq.(2) for convenience, given $t, x_0, c, \theta$. As $n \to \infty$, we have*

$$\sum_{i=1}^{n} \frac{1}{n} L(\epsilon_i) - L \left( \sum_{i=1}^{n} \frac{\sqrt{n}}{n} \epsilon_i \right) < 2K \cdot \sqrt{\frac{2}{\pi}},$$

(31)

*where $\epsilon_i \overset{i.i.d.}{\sim} N(0, I)$ for $i = 1 : n$, $K$ is finite.*

To enable backpropagation, the model loss $L(\epsilon)$ must be differentiable with bounded gradients, satisfying the Lipschitz continuity condition with respect to $\epsilon$. In Corollary A.2, we observe that $\sum_{i=1}^{n} \frac{\sqrt{n}}{n} \epsilon_i \sim N(0, I)$. Instead of computing the entire average, we can directly sample $\epsilon$ from $N(0, I)$ for the forward process. This allows estimating Eq.(18) practically with a single sample $\epsilon$, while ensuring that the estimation error is within a certain range as introduced in Eq.(31).

Please note that this is a very coarse upper bound estimation. In practice, we find that the actual difference is far less than this upper bound. This may be because we use a pretrained large model which has already been well-trained and generalized well to lots of inputs, making the loss typically very close to and small (always within 1 ($K$)). Therefore, we corroborate with experimental results and find that it unnecessary to resample $\epsilon$ repeatedly in the process of optimizing $c_x$.

Moreover, merely comparing the L values is insufficient. We must show that when L values are close, the difference between the corresponding c values is sufficiently small. When the loss difference $L(c_x) - L(c_y)$ is constrained within a certain range, the text embedding difference $c_x - c_y$ will also typically be limited, due to the following reasons:

- Pretrained language generation models are extremely sensitive to even minor changes in input sequences [64], such as the prompt. These subtle alterations can result in significant variations in the model's predicted outcomes and loss function. Therefore, when the difference in loss is constrained within a certain range, we can infer that the semantic disparity in the prompts is also effectively controlled.

Therefore, restricting the loss difference helps bound the text embedding difference, validating that the proposed c values indeed represent meaningful alternative text options rather than arbitrary or randomly perturbed sequences. This ensures the method can generate semantically-meaningful alternatives as expected.

### A.3 Estimation of $H^{-1}$

#### A.3.1 Derivation of $H$

Directly computing the $H$ matrix requires quadratic complexity, and calculating its inverse further requires cubic complexity. Even with an A800 device with 80GB of memory, it is insufficient to support such a computational workload.

Recall that we are interested in estimating the inverse of the Hessian matrix $H = -\nabla\nabla_c \log g(c)|_{c_x}$. In general, the inverse of a matrix is expensive to compute, especially for large matrices. Therefore, we seek to simplify the computation of $H^{-1}$ by leveraging low-order information.

We begin by considering the second-order Taylor expansion of $g(c)$ around $c_x$ (flattened 59,136-dimension vector) :

$$g(c) \approx g(c_x) + \nabla g(c_x)^T (c - c_x) + \frac{1}{2}(c - c_x)^T H(c - c_x) \tag{32}$$

Since $c_x$ is the maximizer of $g(c)$, we have $\nabla g(c_x) = 0$. Therefore, Eq. 32 simplifies to:

$$g(c) \approx g(c_x) - \frac{1}{2}(c - c_x)^T H(c - c_x) \tag{33}$$

We can obtain the estimation formula of $H$ with respect to $L(c)$ since $-L(c)$ is used to compute $g(c)$:

$$H = \nabla\nabla L(c)|_{c_x}, \tag{34}$$

Therefore, we have:

$$(c_0 - c_x)^T H(c_0 - c_x) = 2(L(c_0) - L(c_x)), \tag{35}$$

#### A.3.2 Simplification for Estimation of $H^{-1}$

Research [65] has shown that word embeddings can represent word semantics through distributed, low-dimensional dense vectors. These embeddings capture linguistic patterns and regularities while requiring substantially less memory than sparse high-dimensional representations.

Based on the aforementioned research findings and previous approaches for variance estimation in Laplace modeling [66], we make the assumption that the Hessian matrix is a positive definite diagonal matrix. This assumption is reasonable since calculating the Hessian matrix in the high-dimensional text feature space poses significant computational challenges for stable diffusion models. Furthermore, as the Hessian matrix represents the second-order derivative of the log-likelihood function, it must be positive definite at the maximizer $c_x$ to ensure the local convexity of the function.

**Definition A.3.** *Let* $(c_0 - c_x) = [t_1, t_2, \ldots, t_n]^T$. *Let* $H = diag(h_1, h_2, \ldots, h_n), \frac{1}{l} > h_i > 0, i = 1, 2, \ldots, n$. *Denote* $L := 2(L(c_0) - L(c_x))$. *Define* $D := \sqrt{\sum_{i=1}^{n} \frac{1}{h_i^2}}$.

From Definition A.3, we have:

$$(c_0 - c_x)^T H(c_0 - c_x) = [t_1, t_2, \ldots, t_n] \begin{bmatrix} h_1 & 0 & \ldots & 0 \\ 0 & h_2 & \ldots & 0 \\ \vdots & \vdots & \ddots & \vdots \\ 0 & 0 & \ldots & h_n \end{bmatrix} \begin{bmatrix} t_1 \\ t_2 \\ \vdots \\ t_n \end{bmatrix} = \sum_{i=1}^{n} t_i^2 h_i \tag{36}$$

This implies that Eq.(35) can be written as:

$$\sum_{i=1}^{n} t_i^2 h_i = L \tag{37}$$

Given Eq.(37) as a n-variable linear equation, solving it alone is insufficient. Additional n-1 equations are required to obtain a complete solution. However, due to the large value of n=51,396, solving this system of equations would be highly time-consuming.

To solve this problem, let's consider $\hat{H} = \sigma^2 I$, where $\sigma^2$ is an unknown positive scalar and $I$ is the identity matrix. Replacing $H$ with $\hat{H}$ in Eq.(A.3), we approximate $\sigma^2$ as follows:

$$\sigma^2 = \frac{L}{\sum_{i=1}^{n} t_i^2} = \frac{L}{||c_0 - c_x||_2^2} \tag{38}$$

This implies:

$$\hat{H}^{-1} = (\sigma^2 I)^{-1} = \frac{||c_0 - c_x||_2^2}{L} I \tag{39}$$

On the other hand, in Definition A.3, we assume that the diagonal elements of matrix $H$ are bounded. Since $H$ is a diagonal matrix, its eigenvalues (diagonal elements) are inherently bounded. In our experiments with a learning rate of 0.001 and a step count of 15, we have a bound of $10^{-6}$ for $t_i^2$ from Eq. (37). This leads to an upper bound of $10^6 L$ for $\sum_{i=1}^{n} h_i$, resulting in an average value of the $h_i$ sequence with an approximate upper bound of around $20L$. Based on our observations, we incorporate the following into our standard experimental configuration: $L < 1$, suggesting a value around 20 for $1/l$, and subsequently determine that $D < 12.5$ (We will further explore the implications of different values of $D$ in Figure 4.).

Since Eq. (39) is considered as the final estimation of $H^{-1}$, it is crucial to estimate the upper bound of the distance between $\hat{H}^{-1}$ and $H^{-1}$. As both matrices are high-dimensional and diagonal, we employ the cosine dissimilarity, a widely used metric for measuring the distance between high-dimensional vectors, to quantify the matrix distance. Specifically, we extract the diagonal elements of the matrices as vectors and compute their cosine dissimilarity, which is defined as:

$$\text{Cosine Dissimilarity}(\vec{x}, \vec{y}) = 1 - \frac{\vec{x} \cdot \vec{y}}{||\vec{x}|| ||\vec{y}||} \tag{40}$$

The cosine dissimilarity ranges from 0 to 2, where 0 indicates that the vectors are identical, 1 implies that they are orthogonal, 2 indicates they are maximally dissimilar. We first calculate the cosine similarity:

$$
\begin{aligned}
Similarity &= cos(\hat{H}^{-1}, H^{-1}) \\
&= \frac{\hat{H}^{-1} \cdot H^{-1}}{||\hat{H}^{-1}|| ||H^{-1}||} \\
&= \frac{\sum_{i=1}^{n} \frac{1}{h_i \sigma^2}}{\frac{\sqrt{n}}{\sigma^2} \sqrt{\sum_{i=1}^{n} \frac{1}{h_i^2}}} \\
&= \frac{\frac{1}{n} \sum_{i=1}^{n} \frac{1}{h_i}}{\sqrt{\frac{1}{n} \sum_{i=1}^{n} \frac{1}{h_i^2}}} \\
&\geq \frac{\sqrt{D^2 - (n-1)l^2} + (n-1)l}{\sqrt{n}D},
\end{aligned}
\tag{41}
$$

where $l$ and $D$ are defined in Definition A.3.

We conduct ablation experiments with different $l$ and $D$ values, as shown in Figure 4. Under the standard setting (the yellow point in the bottom left), the cosine dissimilarity between $\hat{H}^{-1}$ and $H^{-1}$ is upper bounded by 0.0909. Even when $D$ increases significantly (close to 60), the cosine dissimilarity does not exceed 0.6.

While this simplification may introduce some errors, it significantly reduces the complexity of computing the Hessian matrix, which is beneficial for model training.

## B  Further Discussion for PAP

We propose to incorporate prompt sampling and optimization into the PGD framework, analogous to AdvDM. Inspired by CW attacks [52], which maps adversarial examples to the tanh space, we can relax the constraints on the optimization problem compared to standard PGD.

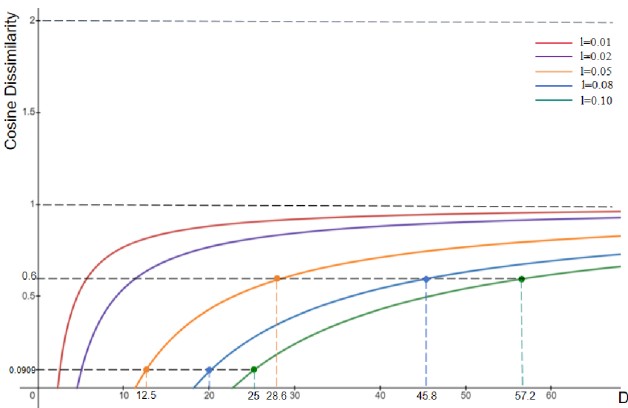

Figure 4: Cosine dissimilarity between $\hat{H}^{-1}$ and $H^{-1}$ under different settings of $D$ and $l$.

Specifically, instead of clipping the perturbations to the $[x_0 - \eta, x_0 + \eta]$ box as in PGD, we can allow the perturbation $\delta$ to vary in the entire space by modifying the image as:

$$x^{\text{adv}} = \frac{1}{2}\big(\tanh(\delta) + 1\big) \tag{42}$$

This has two key advantages: 1.The perturbations $\delta$ are no longer bounded by $\eta$, allowing for more flexible optimization; 2.We can leverage the derivative properties of tanh:

$$\frac{d}{dx}\tanh(x) = 1 - \tanh^2(x) \tag{43}$$

to perform gradient computations without additional cost.

Overall, modifying the space in this way could lead to a smoother optimization process and is a promising direction for future improvement. The full algorithm would involve iteratively optimizing both the prompt embedding and image perturbations via signed gradient steps.

## C  Generalized Prompt-agnostic Adversarial Perturbation

PAP algorithm can incorporate other arbitrary optimizers, such as ASPL, FGSM. The pseudo-code is in Algorithm 3 as an improved algorithm AS-PAP. It is worth noting that when we set the text sampling step to 0, our AS-PAP algorithm is essentially equivalent to Anti-DB [22].

## D  Transfer-based Adversarial Attacks

When crafting adversarial examples, the objective is to find perturbations that maximize the loss function, leading to misclassification or a decrease in the model's confidence. However, the loss landscape is often complex and non-convex, with numerous local optima and saddle points [67]. This landscape structure poses challenges for optimization algorithms, as gradient-based methods can easily get trapped in local optima, resulting in suboptimal or non-transferable adversarial examples.

In the context of our approach to estimate $c_x$, by treating different instances of the loss function $L_t$ as individual "classifiers" mentioned in above works and incorporating momentum in the optimization process, we aim to steer the convergence towards flatter regions in the timestep $t$ landscape. This strategy allows us to find a value of $c_x$ that minimizes the expected value of the integral in Eq. (18) while promoting improved generalization across the sampled loss functions $L_t$.

Analyzing and leveraging the loss landscape in classification adversarial attack opens up new avenues for understanding and improving adversarial robustness. Exhaustive enumeration of all possible perturbations is often computationally expensive and time-consuming. On the other hand, solely fitting the optimization process on a few sampled time steps can lead to overfitting and lack of

**Algorithm 3** Alternating Surrogate and Prompt-agnostic Adversarial Perturbation(AS-PAP)

---

**Input:** images $x_0$, reference prompt $c$, parameter $\theta$, epoch numbers $M$, $N$, $K$, $Max$, learning rates $\alpha$, $r$, $\beta$, $\gamma$, budget $\eta$, noise steps $T$, loss function $L(x, \epsilon, t, c; \theta)$.
**Output:** Adversarial examples $x_M^K$
Initialize $x_0^0 = x_0$
**for** $j = 0$ **to** $N - 1$ **do**
    Sample $t_c \in U(0, T)$, $\epsilon \sim N(0, I)$
    Compute gradient $g_c = \nabla_{c_j} L(x_0, \epsilon_c, t_c, c_j; \theta)$
    Compute momentum $m_{j+1} = \beta m_j + (1 - \beta) g_c$
    Update $c_{j+1} = c_j - r \cdot m_j$
**end for**
**for** $k = 0$ **to** $K$ **do**
    Initialize $x_0^k = x_M^k$, $c_0 = c$
    **for** $i = 0$ **to** $M - 1$ **do**
        Sample $\epsilon, \epsilon_c \sim N(0, I)$, $t \in U(0, T)$
        Compute $c = c_N + \frac{|c_0 - c_N|_2^2}{2 \cdot (L(x_i^k, \epsilon, t, c_0; \theta) - L(x_i^k, \epsilon, t, c_N; \theta))} \cdot \epsilon_c$
        Update $x_{i+1}^k = \text{clip}_{x_0, \eta}(x_i^k + \alpha \cdot \nabla_{x_i^k} L(x_i^k, \epsilon, t, c; \theta))$
    **end for**
    **for** $m = 0$ **to** Max-1 **do**
        $\theta \leftarrow \theta - \gamma \nabla_\theta L(x_M^k, \epsilon, t, c_0; \theta)$
    **end for**
**end for**

---

generalization. It provides insights into the optimization process and offers opportunities to develop more effective and transferable adversarial attacks and defenses.

# E   Implementation Details

## E.1   Additional Details

**Artists Name:** vangogh, john-miler, alfred-sisley, pablo-picasso, abraham

**Test prompt contents:** The specific prompts we use for test on Celeb-HQ, VGGFace2 and Wikiart are shown in Table 7:

|    | Celeb-HQ and VGGFace2 | Wikiart |
|----|------------------------|---------|
| p0 | a photo of sks person. | a sks painting. |
| p1 | a dslr portrait of sks person. | a sks painting of a tree. |
| p2 | an impressionistic depiction of sks person. | a photo of a dog in the style of sks painting. |
| p3 | an abstract representation of sks person. | a photo of a person in the style of sks painting. |
| p4 | a cyberpunk style photo of sks person. | a sks painting of a starry night. |
| p5 | a realistic painting of sks person. | a photo of a lion in the style of sks painting. |
| p6 | a concept art of sks person. | a photo of a sunflower in the style of sks painting. |
| p7 | a headshot photo of sks person. | a photo of a modern building in the style of sks painting. |
| p8 | a caricature sketch of sks person. | a photo of a robot machine in the style of sks painting. |
| p9 | a digital portrait of sks person. | a photo of the Mona Lisa in the style of sks painting. |

Table 7: Test prompts

## E.2 LoRA

LoRA is a method that utilizes low-rank weight updates to improve memory efficiency by decomposing the weight matrix of a pre-trained model into the product of two low-rank matrices, *i.e.*, $W = AB$. This low-rank weight decomposition reduces parameters, minimizing storage needs. While simple, LoRA enhances memory efficiency crucial for large models on constrained hardware. However, defenses alone are limited versus robust methods. LoRA primarily optimizes efficiency over security, so is used to validate stronger defenses against complex attacks. Combining LoRA with such defenses yields balanced, resilient machine learning.

## E.3 Textual Inversion

Textual Inversion allows users to personalize text-to-image generation models with their own unique concepts, without re-training or fine-tuning the model.

The key steps are:

1. Represent a new concept with a pseudo-word $S^*$.
2. Find the embedding vector $v^*$ for this pseudo-word $S^*$ by optimizing the following objective using a small set of images depicting the concept:

$$v^* = \arg\min_v \mathbb{E}_{z \sim E(x), y, \epsilon \sim \mathcal{N}(0,1), t} \left[ \|\epsilon - \epsilon_\theta(z_t, t, c_\theta(y))\|_2^2 \right], \tag{44}$$

where $E$ is the encoder of a pre-trained Latent Diffusion Model (LDM), $x$ are the input images, $y$ are prompts of the form "A photo of $S^*$", $\epsilon_\theta$ is the denoising network, $c_\theta$ is the text encoder, $z_t$ is the noised latent code, and $t$ is the timestep.

3. Use the learned pseudo-word $S^*$ (represented by $v^*$) in natural language prompts to generate customized images with the text-to-image model, *e.g.*, "A painting of $S^*$".

## E.4 Metrics

**CLIP** (Contrastive Language-Image Pretraining) is a framework that not only enables cross-modal understanding between images and text but also allows direct comparison between two images. The CLIP metric measures the similarity between two images using their embeddings.

The CLIP similarity between two images $I_1$ and $I_2$ can be calculated using the following formula:

$$\text{CLIP}(I_1, I_2) = \frac{\text{CosSim}(f(I_1), f(I_2)) + 1}{2} \tag{45}$$

Here, the terms have the following meanings:

$I_1$: the first image, $I_2$: the second image, $f(I_1)$: the embedding of the first image, $f(I_2)$: the embedding of the second image, $\text{CosSim}(x, y)$: the cosine similarity between vectors $x$ and $y$.

The formula calculates the cosine similarity between the embeddings of the two images. The resulting similarity score is normalized to the range [0, 1] by adding 1 and dividing by 2. A higher CLIP value indicates a stronger similarity between the two images. CLIP models are trained on large-scale datasets to learn a joint embedding space for images and text. This enables the models to capture similarities and differences between images using their embeddings. By comparing the embeddings of two images using the cosine similarity, CLIP provides a measure of their visual similarity. The CLIP metric can be used in various tasks such as image retrieval, image similarity search, and image clustering. It allows for effective comparison and organization of images based on their visual content, without the need for explicit labels or annotations.

**LPIPS** (Learned Perceptual Image Patch Similarity) is a metric used to measure the perceptual similarity between two images. It takes into account the local image patches instead of global image features, making it more aligned with human perception. The LPIPS metric is calculated using the following formula:

$$d(x, x_0) = \frac{1}{H_l \cdot W_l} \sum_{h=1}^{H_l} \sum_{w=1}^{W_l} \|w_l \cdot (\hat{y}_{lhw} - \hat{y}_{l0hw})\|_2^2 \tag{46}$$

Here, the terms have the following meanings:

$x$: the input to the model, $x_0$: the reference input, $l$: the layer index, $H_l$: the height of the feature map at layer $l$, $W_l$: the width of the feature map at layer $l$, $\hat{y}lhw$: the predicted output of the model at position $(h, w)$ in layer $l$, $\hat{y}l0hw$: the predicted output of the model at position $(h, w)$ in layer $l$ for the reference input $x_0$, $w_l$: the weight associated with the position $(h, w)$ in layer $l$ The formula calculates the squared Euclidean distance between the weighted differences of predicted outputs at each position $(h, w)$ in layer $l$, normalized by the total number of positions $(H_l \cdot W_l)$. This quantifies the perceptual similarity between the input $x$ and the reference input $x_0$ at the specified layer. LPIPS provides a perceptually meaningful measure for comparing images, capturing both local and global information. It has been widely used in various computer vision tasks and can be especially useful for evaluating the performance of image synthesis models.

**BRISQUE** is a widely used no-reference image quality assessment metric that evaluates the quality of an image without relying on a reference image. It computes a quality score based on statistical features extracted from the image, such as brightness, contrast, and naturalness. The BRISQUE score ranges from 0 to 100, with higher scores indicating better image quality. The computation involves:

1. Extracting local normalized luminance statistics from the image.

2. Computing a feature vector from the statistics using a pre-trained model.

3. Mapping the feature vector to a quality score using a support vector regression model.

BRISQUE is effective in capturing distortions introduced by various image processing operations and is widely used in benchmarking image generation models.

**LAION aesthetics predictor** is a linear model that takes CLIP image encodings as input. It was trained on a dataset of 17,600 images rated by humans. The images in the training set were scored on a scale from 1 to 10, with higher scores typically indicating artistic quality. In subsequent experiments, the LAION aesthetics predictor assigns scores to samples generated by diffusion models. Higher scores indicate a higher level of artistic quality in the images.

### E.5 Baselines

To ensure a fair comparison, we followed the original settings of AdvDM, IadvDM, and Anti-Dreambooth in our experiments. We constrained the noise budget to be the same for all methods and focused on comparing their performance in untargted scenarios. This approach avoids the bias introduced by selecting specific target images, as each method may have its own optimized target image. Hence, we standardized the evaluation by considering untargted scenarios. For the attack stage, we set the number of steps to 20. If a method involved training Dreambooth, we limited the training steps to 10. To expedite the training process, we used a batch size of 20 for training Dreambooth.

## F Additional Study

### F.1 More evidence of protecting fail of previous methods

The Table 8 presents the results of previous protective perturbation methods (No defense, AdvDM, and Anti-DB) on the CelebA-HQ dataset, evaluated using the BRISQUE (higher is better) and CLIP (lower is better) metrics. The results are reported for ten different test prompts, where p0 is the same as the training prompt.

For the training prompt p0, both AdvDM and Anti-DB outperform the "No defense" baseline, achieving higher BRISQUE scores (39.85 and 40.08 vs. 21.20) and lower CLIP scores (0.2053 and 0.1877 vs. 0.4821). This indicates that the protective perturbations are effective when the test prompt matches the training prompt.

However, for the other test prompts (p1 to p9), which differ from the training prompt, the performance of AdvDM and Anti-DB is not consistently better than the "No defense" baseline. For instance, considering the BRISQUE metric, AdvDM achieves similar scores to "No defense" for prompts p2 (29.45 vs. 27.65), p4 (29.78 vs. 28.45), and p7 (26.89 vs. 26.73). Anti-DB shows similar BRISQUE

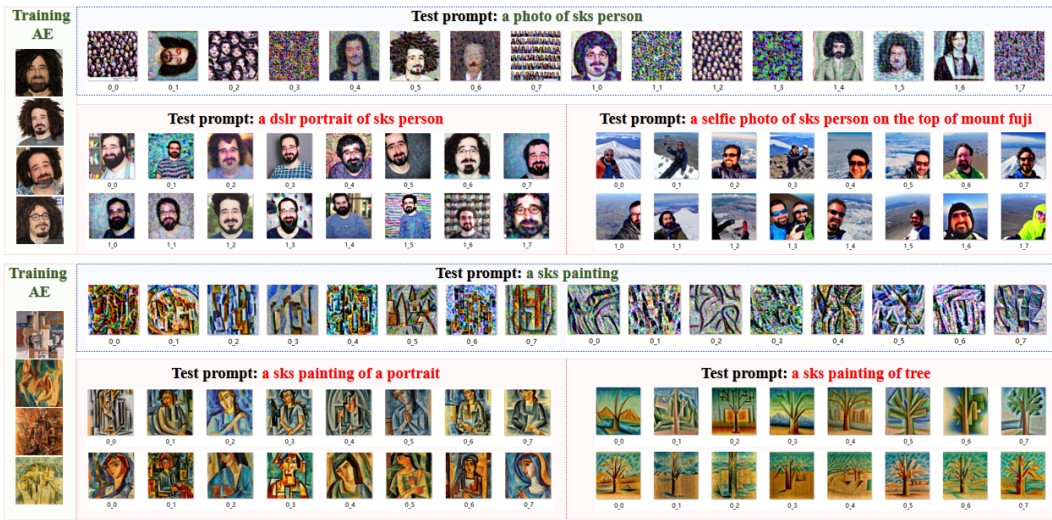

Figure 5: Visualized results of different test prompts toward Anti-DB method on the CelebA-HQ dataset and Wikiart dataset, with training prompt: a photo of sks person (top) / a sks painting (bottom). The left column are adversarial examples (denoted as AE) by Anti-DB.

scores to "No defense" for prompts p3 (33.14 vs. 29.11), p5 (31.51 vs. 30.22), and p7 (31.74 vs. 26.73).

The CLIP metric also exhibits a similar trend, where AdvDM and Anti-DB do not consistently outperform "No defense" for prompts different from the training prompt. For example, AdvDM has similar CLIP scores to "No defense" for prompts p2 (0.3124 vs. 0.2543), p4 (0.3267 vs. 0.2987), p7 (0.2654 vs. 0.3015), p8 (0.3012 vs. 0.2976), p9 (0.3129 vs. 0.3211) indicating poor defense performance.

These observations suggest that while AdvDM and Anti-DB are effective in protecting against the training prompt, their performance deteriorates when the test prompt deviates from the training prompt. This highlights the challenge of developing robust protective perturbations that can generalize to unseen prompts, which is a crucial requirement in real-world scenarios.

Table 8: Results of previous protective perturbations using ten different test promtps (Prompt contents in Table 7 of Appendix) on the CelebA-HQ dataset. p0 is the same as the training prompt.

| Metrics | Method | p0 | p1 | p2 | p3 | p4 | p5 | p6 | p7 | p8 | p9 |
|---|---|---|---|---|---|---|---|---|---|---|---|
| BRISQUE (↑) | No defense | **21.20** | 25.34 | 27.65 | 29.11 | 28.45 | 30.22 | 27.89 | 26.73 | 31.05 | 28.98 |
| | AdvDM | **39.85** | 37.12 | 29.45 | 35.67 | 29.78 | 33.14 | 37.25 | 26.89 | 36.52 | 32.23 |
| | Anti-DB | **40.08** | 38.96 | 36.65 | 33.14 | 34.98 | 31.51 | 32.82 | 31.74 | 37.66 | 35.98 |
| CLIP (↓) | No defense | **0.4821** | 0.4675 | 0.2543 | 0.4032 | 0.2987 | 0.3465 | 0.4211 | 0.2654 | 0.3012 | 0.3129 |
| | AdvDM | **0.2053** | 0.2687 | 0.3124 | 0.2845 | 0.3267 | 0.2895 | 0.2532 | 0.3015 | 0.2976 | 0.3211 |
| | Anti-DB | **0.1877** | 0.2612 | 0.3029 | 0.2754 | 0.3165 | 0.2806 | 0.2459 | 0.2924 | 0.2884 | 0.3115 |

Additionally, from Figure 5, when the test prompt matches the training prompt (*e.g.* green prompt, the test images exhibit stable interference. However, when the test prompts differ, the test images maintain high quality and semantic consistency with test prompts, nearly resembling scenarios without any defense. This failure in previous protection is evident from the red prompt-generated images in Figure 5

### F.2 Sampling steps for training DreamBooth

The selection of the number of steps for fine-tuning is also a critical factor that affects the quality of the generated images. Therefore, we conducted ablation experiments with different sampling step values: 500, 1000, 1200, 1600, and 2000. The results in Table 9 demonstrate the stable defense effectiveness of our method across different sampling test step settings.

Table 9: Ablation study of sampling test steps.

| Sampling steps | FID ($\uparrow$) | CLIP-I ($\downarrow$) | LPIPS ($\uparrow$) | LAION ($\downarrow$) | BRISQUE ($\uparrow$) | CLIP ($\downarrow$) |
|---|---|---|---|---|---|---|
| 500 | **452.0** | **0.5467** | 0.7363 | **5.293** | **38.91** | 0.2875 |
| 1000 | 448.3 | 0.5641 | 0.7782 | 5.490 | 38.47 | 0.2654 |
| 1200 | 437.9 | 0.5909 | **0.7889** | 5.419 | 38.12 | 0.2387 |
| 1600 | 446.8 | 0.5948 | 0.7683 | 5.587 | 37.98 | 0.2032 |
| 2000 | 433.2 | 0.5978 | 0.7759 | 5.338 | 37.24 | **0.1921** |

## F.3 Sampling steps for $\epsilon$ sampling steps

We also analyze the effect of $\epsilon$ sampling steps used to estimate $c_x$, which is approximated in our method. As shown in Table 10, metrics such as FID, CLIP, and LPIPS peak at 10 $\epsilon$ steps, then stabilize and resemble the 0 $\epsilon$ step results from 15 steps onward. This validates our method for approximating $c_x$ in Eq.(9).

Table 10: Ablation study of $\epsilon$ sampling steps on Wikiart.

| $\epsilon$ sampling steps | FID ($\uparrow$) | CLIP-I ($\downarrow$) | LPIPS ($\uparrow$) |
|---|---|---|---|
| 0 | 448.3 | 0.5641 | 0.7782 |
| 5 | 446.8 | 0.5607 | 0.7743 |
| 10 | 482.2 | 0.5598 | 0.7801 |
| 15 | 454.5 | 0.5639 | 0.7788 |
| 20 | 453.9 | 0.5602 | 0.7775 |

## F.4 Noise budget

Table 11 shows the impact of the noise budget $\eta$ on PAP's defense performance. The trade-off between noise stealthiness and defense performance is important to consider. A noise budget of $\eta = 0.05$ is effective, but increasing the budget improves performance at the expense of stealth.

We provide a detailed analysis of the impact of noise budget on our experimental results, with additional visualizations available in Figure I. In general, a larger noise budget yields better defense performance. However, it also introduces more noticeable image distortions. In extreme cases, excessive perturbations can degrade the image to pure noise, defeating the purpose of our task. Hence, striking a balance between protection effectiveness and the magnitude of image perturbations becomes essential.

## F.5 Other Pseudo-word

In our experiments, we conducted evaluations using the commonly used pseudo-word "sks," which is representative but may not cover all possible cases. To further validate our method, we included additional less commonly used pseudo-words. We selected three traditional metrics for measuring image similarity, and the results in Table 12 clearly demonstrates that our method consistently outperforms the others.

## F.6 Robustness

Adversarial examples often lose their protective effect on images when subjected to image operations such as Gaussian blur, JPEG compression, etc. Therefore, we conduct robustness tests specifically targeting the JPEG compression and Gaussian Blur. We evaluate the quality of the generated images at different settings, as shown in Table 13. Despite a slight decline in our results after applying these treatments, it is worth noting that the evaluation based on image quality metrics still demonstrates favorable outcomes. This suggests that our method maintains a good level of effectiveness even in the presence of JPEG compression and Gaussian Blur.

## F.7 Targeted Attack

Targeted attack on generative models aims to perturb an input image towards a specific target image, resulting in outputs that closely resemble the target. Compared to untargeted attack, targeted attack

Table 11: Ablation study of noise budget, ranging from 0.01 to 0.15.

| $\eta$ | FID ($\uparrow$) | CLIP-I ($\downarrow$) | LPIPS ($\uparrow$) | LAION ($\downarrow$) | BRISQUE ($\uparrow$) | CLIP ($\downarrow$) |
|---|---|---|---|---|---|---|
| - | 198.7 | 0.7715 | 0.6193 | 6.367 | 27.34 | 0.3515 |
| 0.01 | 262.7 | 0.7188 | 0.6522 | 5.994 | 31.12 | 0.3154 |
| 0.03 | 334.8 | 0.6632 | 0.6840 | 5.603 | 35.57 | 0.2896 |
| 0.05 | 448.3 | 0.5641 | 0.7782 | 5.490 | 38.47 | 0.2654 |
| 0.10 | 498.5 | 0.4914 | 0.8384 | 5.003 | 42.22 | 0.2088 |
| 0.15 | **512.9** | **0.4208** | **0.8755** | **4.635** | **46.26** | **0.1716** |

Table 12: Comparison with other adversarial attack methods with pseudo-word "t@t"

| Dataset | Method | FID ($\uparrow$) | CLIP-I ($\downarrow$) | LPIPS ($\uparrow$) |
|---|---|---|---|---|
| Celeb-HQ | Clean | 142.3 | 0.7872 | 0.4725 |
| | AdvDM | 187.8 | 0.7023 | 0.5213 |
| | Anti-DB | 197.4 | 0.6961 | 0.5597 |
| | IAdvDM | 161.6 | 0.7488 | 0.4912 |
| | PAP (Ours) | **228.6** | **0.5749** | **0.6348** |
| VGGFace2 | Clean | 239.7 | 0.6504 | 0.5529 |
| | AdvDM | 240.6 | 0.6432 | 0.5733 |
| | Anti-DB | 254.4 | 0.6307 | 0.6097 |
| | IAdvDM | 244.9 | 0.6336 | 0.5955 |
| | PAP (Ours) | **272.8** | **0.5301** | **0.6877** |
| Wikiart | Clean | 177.7 | 0.7339 | 0.5929 |
| | AdvDM | 313.7 | 0.6704 | 0.6558 |
| | Anti-DB | 349.2 | 0.6489 | 0.6910 |
| | IAdvDM | 302.8 | 0.6641 | 0.6590 |
| | PAP (Ours) | **383.0** | **0.6167** | **0.7168** |

achieves more consistent and effective results. We conduct additional experiments. The results are presented in Table 14.

## F.8 The Performance of the Trained Prompts

In Table 15, our method slightly trails behind the SOTA by 0.01/39.57 in LPIPS/FID respectively. However, our method still maintains a leading position in ISM, FDFR, BRISQUE, and CLIP metrics.

## F.9 Time Comparisons with Baselines

In Table 16, we present the time required for each method. The results demonstrate that PAP introduces an average computation time of processing a set of images (<300s). In the future, we aim to further optimize the algorithm to reduce the time to within 4 minutes.

## F.10 Error Bars

Due to the sampling from three different distributions involved in our algorithm, the results obtained in each experimental run may exhibit slight variations. To address this, we conduct multiple independent repetitions of the experiments and calculate the mean and standard deviation for each evaluated metric. The results are then presented with error bars, providing a more comprehensive assessment of the algorithm's performance.

Specifically, for each metric to be evaluated, we perform $N = 10$ independent experiment repetitions, obtaining $N$ result values $x_1, x_2, \ldots, x_N$. The mean $\mu$ and standard deviation $\sigma$ of these values are calculated as follows:

$$\mu = \frac{1}{N} \sum_{i=1}^{N} x_i \tag{47}$$

$$\sigma = \sqrt{\frac{1}{N-1} \sum_{i=1}^{N} (x_i - \mu)^2} \tag{48}$$

Table 13: Metrics for images generated by PAP after JPEG compression or Gaussian Blur.

| | LAION ($\downarrow$) | BRISQUE ($\uparrow$) |
|---|---|---|
| PAP | 5.490 | 38.47 |
| JPEG Comp. Q=10 | 5.732 | **45.33** |
| JPEG Comp. Q=30 | **5.485** | 34.29 |
| JPEG Comp. Q=50 | 5.561 | 33.63 |
| JPEG Comp. Q=70 | 5.593 | 39.34 |
| Gaussian Blur K=3 | 5.882 | 41.88 |
| Gaussian Blur K=5 | 5.796 | 42.08 |
| Gaussian Blur K=7 | 5.552 | 43.76 |
| Gaussian Blur K=9 | 5.991 | 41.20 |

Table 14: Ablation study of targeted attack.

| - | FID ($\uparrow$) | CLIP-I ($\downarrow$) | LPIPS ($\uparrow$) | LAION ($\downarrow$) | BRISQUE ($\uparrow$) | CLIP ($\downarrow$) |
|---|---|---|---|---|---|---|
| untarget | 448.3 | 0.5641 | 0.7782 | 5.490 | 38.47 | 0.2654 |
| target | **492.1** | **0.5335** | **0.7922** | **4.974** | **40.23** | **0.2108** |

When plotting the results, we use the mean $\mu$ as the metric's result value and the values $\mu \pm \sigma$ as the upper and lower limits of the error bars, respectively. By presenting the error bars, we aim to provide a more reliable and informative evaluation of our proposed method.

As shown in Figure 6, the error bars of our PAP method demonstrate the stability and consistency of our method's performance across different runs.

# G   Impact Statements

Powerful customized diffusion models enable many applications but also raise privacy and intellectual property concerns [68, 19] if misused to reconstruct private images or replicate protected artistic works without consent [16, 69]. Previous work on privacy protection has been questioned in its effectiveness [70, 71] and suffers from a lack of generalization to unknown prompts [72]. Our proposed Prompt-agnostic Adversarial Perturbation technique aims to address the limitations by considering the distribution of text prompts the attacker may use. Experiments demonstrate our approach more effectively prevents unauthorized use of sensitive private data and artistic styles compared to state-of-the-art baselines under unseen prompts. By enhancing resilience against unseen attacks targeting uncontrolled generation, our work can help balance AI's societal benefits with mitigating privacy and legal risks, and provides insights towards continued research on defense techniques that responsibly enable trustworthy applications involving generation of human data.

However, our model could also be used to maliciously contaminate data sources. If adversarial examples generated by our model are mixed with regular natural images, it could potentially contaminate the entire dataset, leading to its abandonment, as the adversarial perturbations we introduce may be difficult to detect.

# H   Limitations and Future Work

## H.1   Semantic Relevance in Prompt Sampling

While modeling the prompt distribution as a continuous Gaussian distribution is a reasonable approximation, we must acknowledge that the sampled c values may not always reflect realistic semantics in the text feature space. For instance, simply adding noise to text embedding can result in incoherent sentences that lack real-world meaning. This poses a challenge to the effectiveness of our global protection when sampling from the prompt distribution. In reality, we aim to consider prompt samples that are both close to the mean and contain meaningful semantic information, resembling a discrete scenario.

To further explore the relationship between the estimated prompts and natural language, we have conducted a visual experiment in Figure 7. By reducing the dimensionality of 10 test prompts' embeddings and $c_N$ using PCA, we visualize the estimated prompt distribution and test prompts projected in a low-dimensional space for the tasks of "Facial Protection" (left) and "Preservation of Artistic Style" (right) respectively. The figures demonstrate that the two principal components of

Table 15: Performance of the trained prompts.

| Method | LPIPS | FDFR | ISM | BRISQUE | FID | CLIP |
|---|---|---|---|---|---|---|
| AdvDM | 0.65 | 0.61 | 0.39 | 33.68 | 301.55 | 0.25 |
| Anti-DB | 0.71 | 0.68 | 0.34 | 32.24 | 277.24 | 0.24 |
| IAdvDM | 0.65 | 0.57 | 0.43 | 33.55 | 296.31 | 0.28 |
| PAP | 0.70 | 0.68 | 0.33 | 36.43 | 261.98 | 0.24 |
| No Defense | 0.50 | 0.01 | 0.55 | 23.22 | 128.31 | 0.38 |

Table 16: Time comparisons with baselines.

| Method | AdvDM | Anti-DB | IAdvDM | PAP |
|---|---|---|---|---|
| Time | 262s | 288s | 204s | 297s |

test prompts are discretely distributed within the modeled prompt distribution, indicating a flexible probability of being selected for adversarial attacks. This illustrates that our modeling effectively covers a range of natural language inputs in the semantic space.

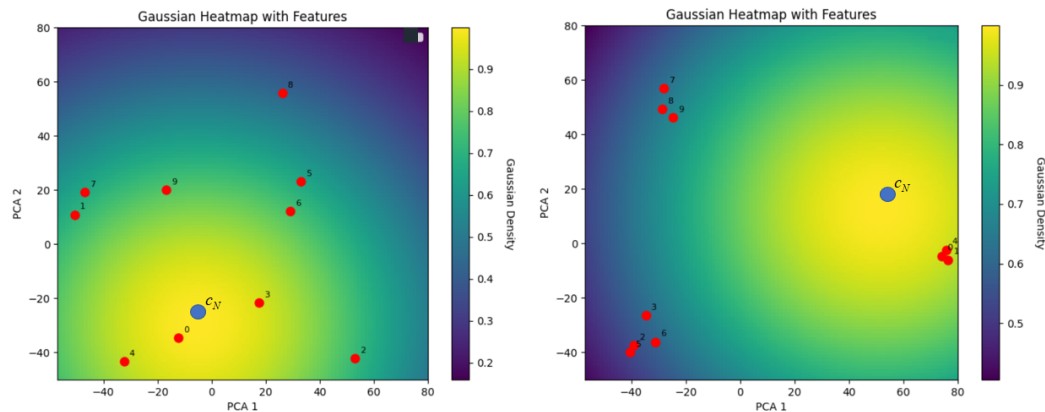

Figure 7: Evaluation of generation performance of models across different stable diffusion versions on different metrics.

In future work, one potential improvement is to discretize the modeled Gaussian distribution by introducing a restriction module, denoted as F. This module would reject prompt samples lacking semantic information, allowing only those with strong semantic relevance to enter the optimization process. This approach would provide a more accurate representation of meaningful alternative text options.

## H.2  $H^{-1}$ Estimation

In estimating $H^{-1}$, we made simplifications that inevitably introduced errors. Although we theoretically demonstrated that the upper bound of the error caused by our simplification is sufficiently small, it is worth exploring if there are alternative estimation methods that can achieve even smaller and more accurate errors.

In future work, we may consider other estimation approaches for $H^{-1}$, such as matrix low-rank decomposition, to pursue a more rigorous approximation. By exploring these alternative methods, we aim to discover more precise and reasonable estimations for $H^{-1}$.

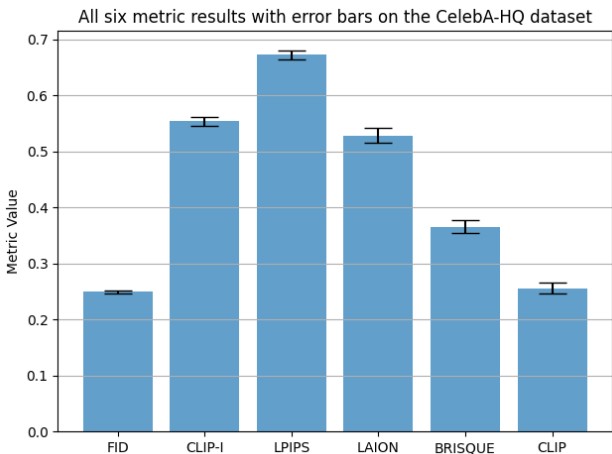

Figure 6: All six metric results with error bars on the CelebA-HQ dataset. The error bars indicate the stability of the results across multiple runs. For the purpose of visual representation, the results of FID, BRISQUE, and LAION in the graph are obtained by multiplying the original data by 0.001, 0.01, and 0.1, respectively.

# I  Visualization

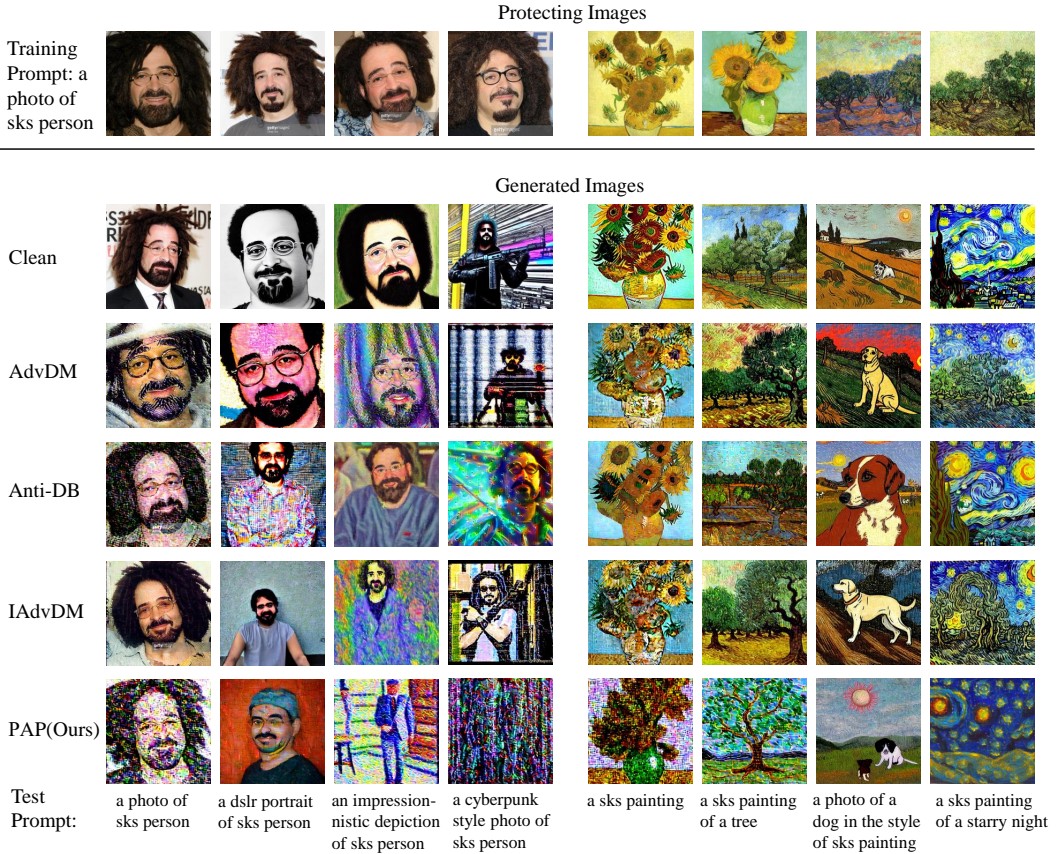

Figure 8: Clean examples and generated images (qualitative defense results) of different methods in VGGFace2 (left) and Wikiart (right). Each row represents a method, and each column represents a different test prompt (shown at the bottom). The adversarial examples generated by our method effectively defend against all prompts in both datasets. In contrast, other comparison methods primarily focus on protecting the fixed prompt (the first column), resulting in compromised defense for other prompts.

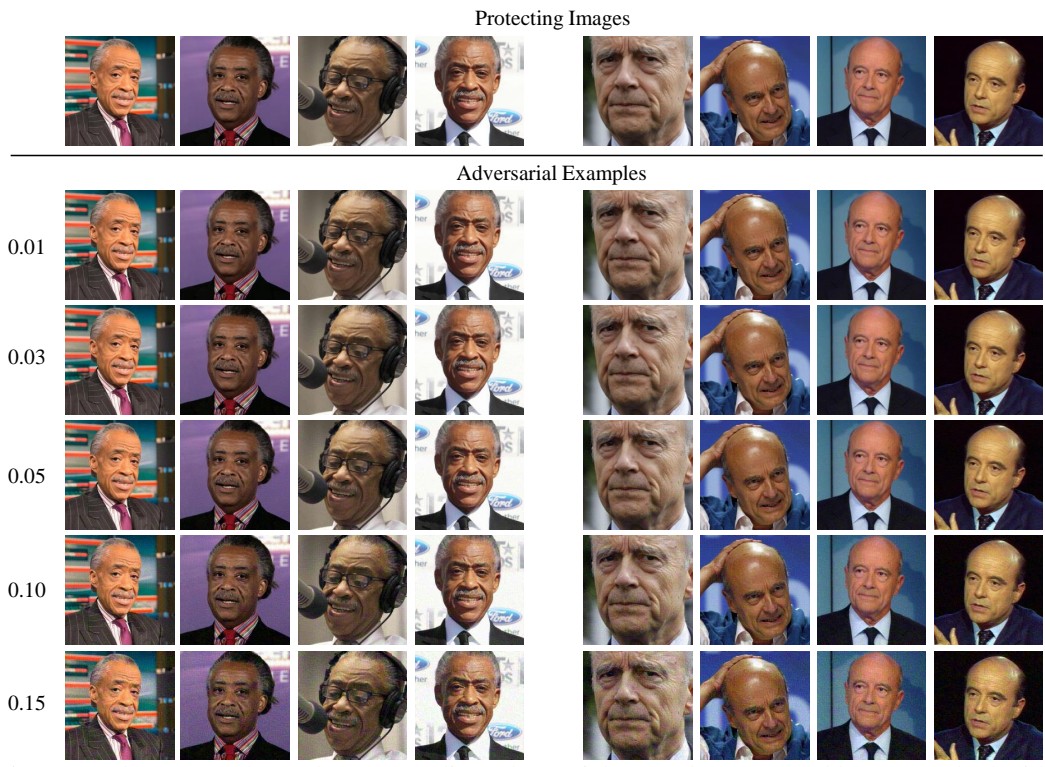

Figure 9: Clean examples and corresponding adversarial examples generated by our method with different noise budgets (ranging from 0.01 to 0.15) on VGGFace2. The training prompt is "a photo of sks person".

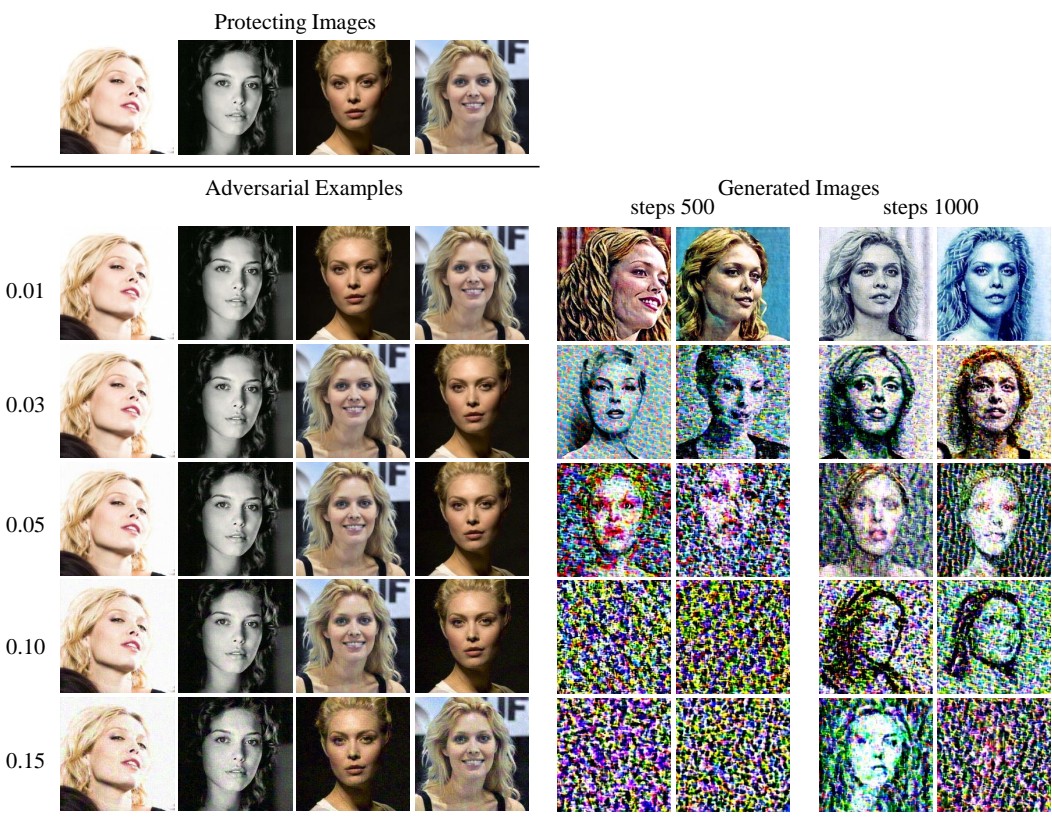

Figure 10: Top: clean examples. Bottom left: adversarial examples produced by our method with different noise budgets (ranging from 0.01 to 0.15) on VGGFace2 with training prompt: "a photo of sks person". Bottom right: corresponding generated images of stable diffusion with sampling steps 500 and sampling steps 1,000.

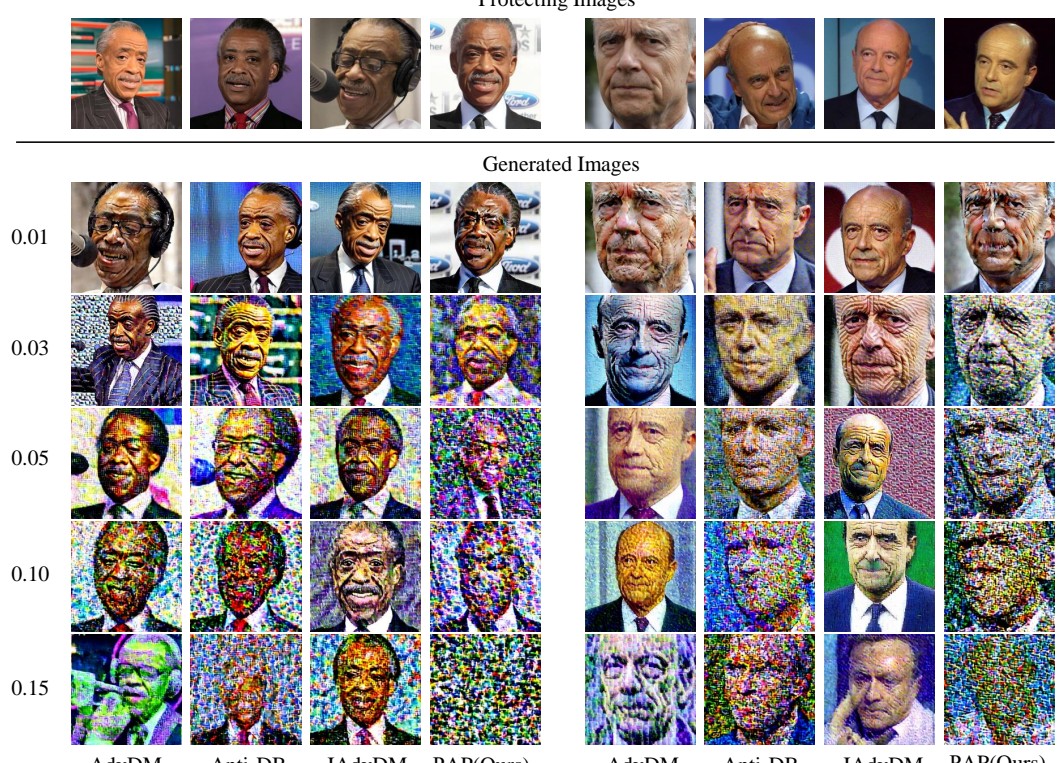

Figure 11: Top: clean examples. Bottom: generated images of different methods with different noise budgets (ranging from 0.01 to 0.15) on VGGFace2. The training prompt is "a photo of sks person".

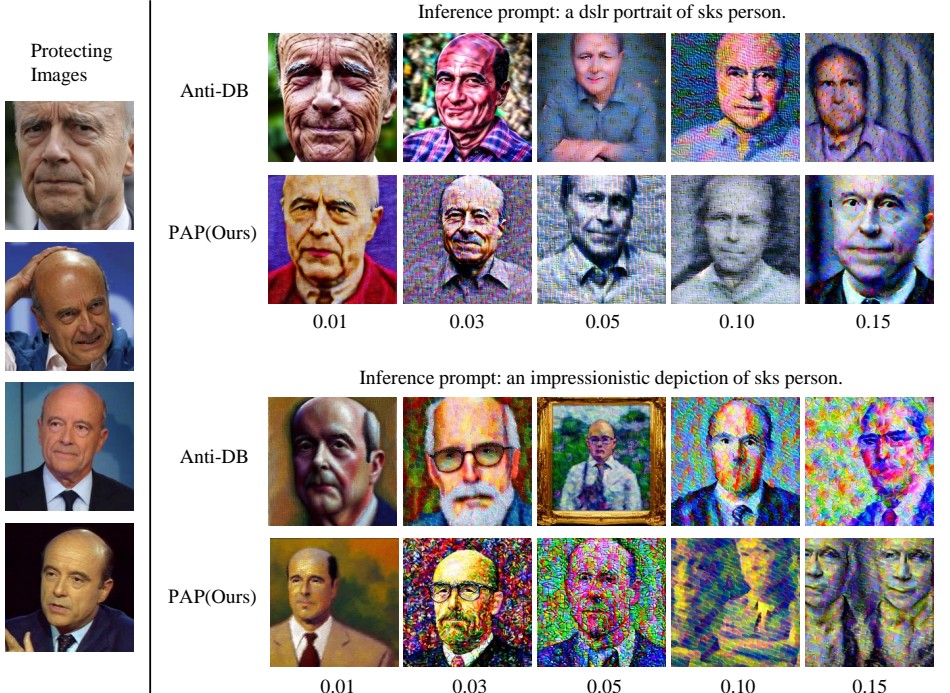

Figure 12: Left: clean examples. Top right: generated images of Anti-DB and our method with different noise budgets (ranging from 0.01 to 0.15) on VGGFace2. The inference prompt is "a dslr portrait of sks person". Bottom right: generated images of Anti-DB and our method with different noise budgets on VGGFace2. The inference prompt is "an impressionistic depiction of sks person".

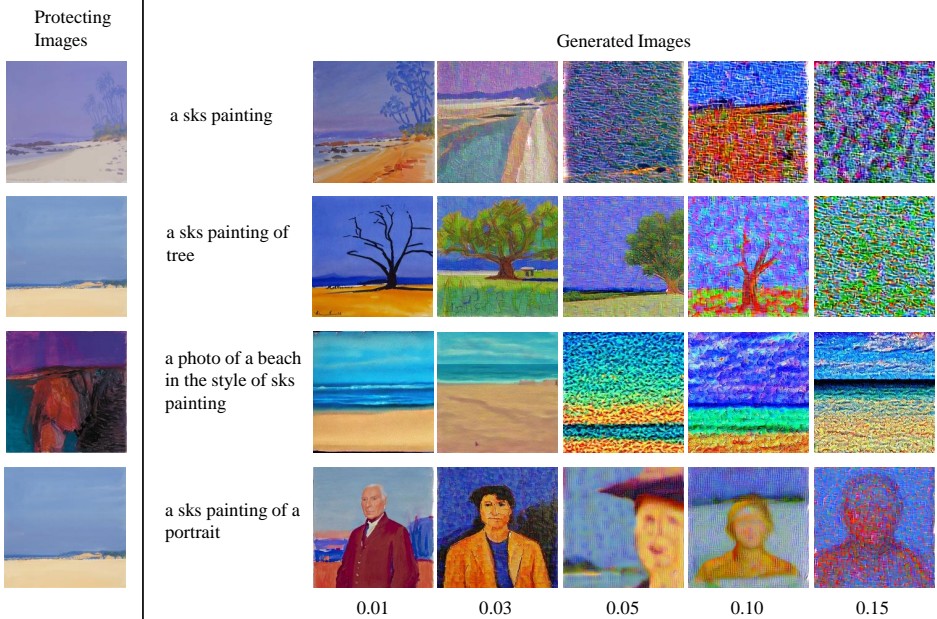

Figure 13: Left: clean examples. Right: generated images of our method with different noise budgets (ranging from 0.01 to 0.15) and prompts on Wikiart.

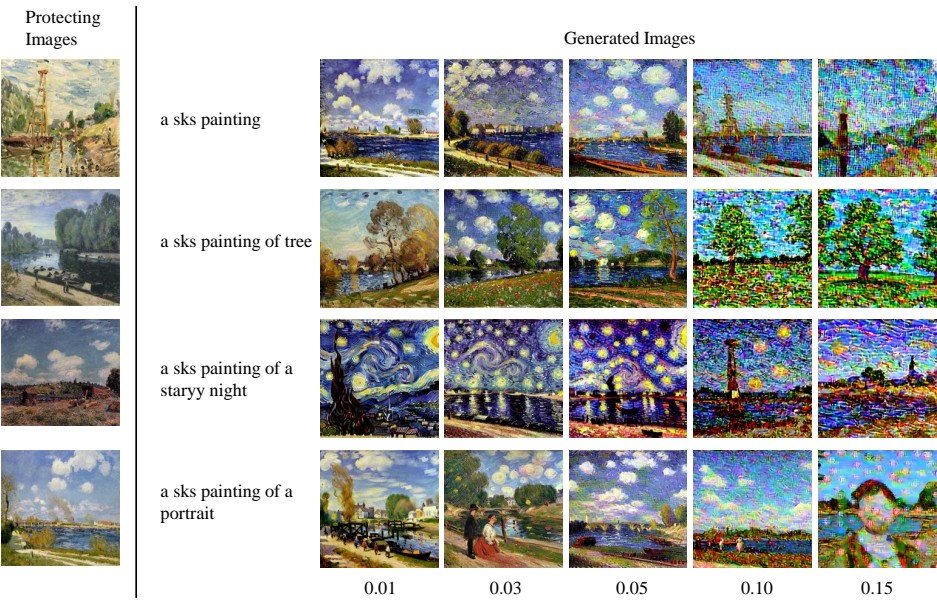

Figure 14: Left: clean examples. Right: generated images of our method with different noise budgets (ranging from 0.01 to 0.15) and prompts on Wikiart.

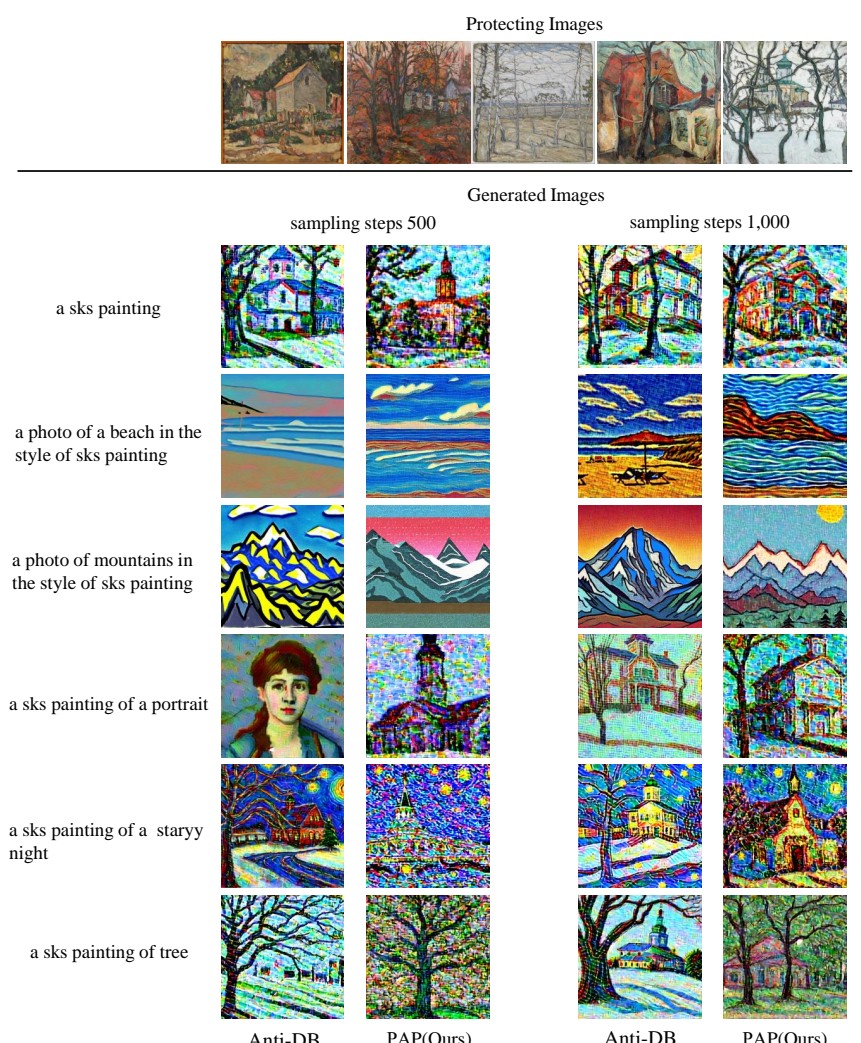

Figure 15: Top: clean examples. Bottom: generated images of our method with different prompts and sampling steps (500 and 1,000) on Wikiart.

