# OpenReview forum: "Prompt-Agnostic Adversarial Perturbation for Customized Diffusion Models"
_NeurIPS.cc/2024/Conference — NeurIPS 2024 poster_

### Official Review · Reviewer_Q3Lh · 2024-06-19

**Soundness:** 3
**Presentation:** 4
**Contribution:** 3
**Rating:** 5
**Confidence:** 4

**Summary:**

The paper presents Prompt-Agnostic Adversarial Perturbation (PAP), a method to enhance privacy and security in customized text-to-image diffusion models like Stable Diffusion.  These models, while enabling high-quality image synthesis, pose risks of privacy breaches and unauthorized artwork usage.  Existing adversarial methods, limited by their reliance on specific prompts, fail with unseen prompts.  PAP addresses this by modeling the prompt distribution with a Laplace approximation, generating robust perturbations effective against diverse prompts. Experiments on datasets such as VGGFace2 and Wikiart show PAP's superior performance and robustness.  This method provides a significant improvement in protecting images from unauthorized use and manipulation in diffusion models.

**Strengths:**

1. I appreciate the idea of designing a prompt-agnostic adversarial perturbation for privacy protection.
2. The method is concise and paper organization is logical.
3. The formulas, tables, and figures in the paper are easy to follow.

**Weaknesses:**

1. Although the motivation for designing a prompt-agnostic adversarial perturbation is reasonable, the method seems too complex. That is, maybe we can expand the prompt with LLM to get a lot of prompts (different scenes) containing the keyword. Then we can use all these prompts together as a set to calculate a gradient for adversarial attack at each iteration.
2. For definition 3.1, the assumption of Z = p(x0, c0) as a constant lacks evidence.

Minor suggestion:

1.	For tables 1, 2, 3, and 4, there are some left parentheses that have no space between the previous metric names.

**Questions:**

Please see weakness

---

> ### Author Rebuttal · Authors · 2024-08-06
>
> Dear Reviewer Q3Lh,
>
> Thank you for your thorough review and constructive feedback. Your perceptive comments and suggestions have helped us improve our work.
>
> Q1: a) PAP seems too complex? b) Simply expand prompts with LLM and gradient ensemble?
> > a) The final implementation of PAP, as presented in **Algorithm 1**, is notably straightforward and exceptionally user-friendly.
> The pipeline of the entire model involves initially modeling a prompt distribution through Laplace approximation, with the development of two estimators to compute the distribution parameters. Subsequently, Monte Carlo sampling is applied to each input distribution to optimize a disturbance expectation. The expressions for the two estimators are depicted in Equations (9) and (12) in the paper. On this basis, our implementation merely entails sampling attacks on Gaussian distributions with means from Equations (9) and variances from Equation (12), as Reviewer JLDf remarked, **"PAP is simple and easy to use."**
> In order to further expound on the precision and interpretability of our estimation methods, we rigorously assess the errors between the two estimators and the ground truth, and proceed with deriving upper bounds for these errors in Appendix A. Hyperparameters are meticulously selected to ensure that these error bounds stay within manageable limits. While the derivation of this error assessment may have initially appeared complex, these efforts are aimed at streamlining the final algorithm's implementation. Through above detailed mathematical reasoning, we have arrived at a **concise** algorithm and implementation code outlined in **Algorithm 1**, which stands as a significant highlight of our work.
>
> > b) The approach you mentioned involves expanding the prompt using a Large Language Model (LLM) to generate multiple prompts (different scenes) containing the keyword. Subsequently, all these prompts are utilized collectively as a set to calculate the gradient for the adversarial attack at each iteration.
> This gradient ensemble approach relies on a crucial **assumption**: through a straightforward ensemble technique like averaging the gradients of loss concerning all $g_i$ during the adversarial attack process, one can obtain an ensembled $g$ that maximizes the overall loss for each prompt. However, this fundamental premise has faced criticism for "often overlooking the unique attributes of each model, resulting in suboptimal outcomes" **[1]**, making it challenging to achieve the desired optimal results.
>  Contrary to this simple ensemble strategy, by modeling the prompt distribution and conducting targeted attacks based on the probabilistic sampling within the modeled distribution, our method accounts for the **diverse characteristics** of each prompt and their respective influences on the overall attack, thus leading to a more adaptive and effective adversarial strategy.
>
> Q2: The assumption of $Z$ as a constant in definition 3.1
> > In Definition 3.1, $x_0$ represents the given input image, while $c_0$ represents the given descriptive text. Both are given inputs rather than variables, and as such, $Z = p(x_0, c_0)$ is a constant.
>
> Q3: For tables 1, 2, 3, and 4, there are some left parentheses that have no space between the previous metric names.
> > We appreciate your kind reminder. If there are any additional suggestions or concerns, please feel free to share them with us.
>
> We sincerely appreciate your considerate review and the insightful feedback you provided for our paper. It seems that you have viewed our work favorably, for which we are grateful. Would you kindly consider adjusting the rating to better align with your positive sentiments towards our research? Your understanding and support mean a great deal to us.
>
> [1]An Adaptive Model Ensemble Adversarial Attack for Boosting Adversarial Transferability

---

### Official Review · Reviewer_nNzN · 2024-07-11

**Soundness:** 3
**Presentation:** 3
**Contribution:** 3
**Rating:** 7
**Confidence:** 3

**Summary:**

This paper proposes a novel adversarial training method for text-to-image diffusion models, enhancing robustness against prompt-agnostic attacks. Specifically, the authors utilize prompts (embeddings) from a prompt distribution rather than a specific prompt for adversarial training. The proposed method is evaluated in the contexts of face privacy and artistic style protection.

**Strengths:**

1. This paper aims to bridge the gap between prompt-specific protection and prompt-agnostic protection, which is essential for text-to-image diffusion models.
2. The proposed method is principled and effective.
3. The empirical evaluation is comprehensive and convincing.

**Weaknesses:**

See Questions.

**Questions:**

1. Is the proposed protection method robust against adversarial prompts?
2. Is it possible to demonstrate the connection between the test prompt and the prompt distribution used for training? Alternatively, can sampled embeddings from the prompt distribution be projected into natural language? Such studies would be beneficial for illustrating that the constructed prompt distribution adequately covers potential user prompts.

**Limitations:**

The authors discussed the limitations in Appendix H.

---

> ### Author Rebuttal · Authors · 2024-08-06
>
> Dear Reviewer nNzN,
>
> We appreciate the time and effort you put into providing feedback on our work. Your insightful comments have contributed to the enhancement of our paper.
>
> Q1: Demonstrate the connection between the test prompt and the prompt distribution
> > We demonstrate the robustness of PAP against adversarial prompts from two perspectives:
>
> > a) **Theoretical Perspective**:  Unlike previous approaches, we target the entire prompt distribution. Optimizing adversarial perturbations in this manner yields effective results even when faced with unknown prompts, thus achieving a more robust attack effectiveness against different adversarial prompts. Our modeling of prompt distribution is based on Laplace approximation.  Subsequently, we offer a detailed derivation, encompassing the Laplace modeling distribution and introducing two estimators for mean and variance estimation, along with an estimation of upper bounds for errors in Appendix A. A series of derivations demonstrate that our estimations fall within controllable margins.
>
> > b) **Experimental Perspective**:  In the paper, Table 1 showcases the average results of PAP across three datasets with 10 different test prompts each. Figure 2 displays visual representations of the results for different test prompt inputs, while Figure 3 exhibits the outcomes for combined inputs of different test prompts. Table 10 reveals the results of test prompts with other pseudo words used as inputs. Additionally, Appendix I presents further visual results of different training and test prompts. Collectively, these results indicate that PAP is robust against adversarial prompts.
>
> Q2: Connection between the test prompt and the prompt distribution
> >  Thank you very much for your constructive feedback.
> As emphasized in Appendix H.1, selecting prompt samples that approximate the mean and convey meaningful semantic information is vital for bridging the semantic gap between the estimated prompt distribution and natural language. In our ongoing efforts, we propose a restriction module to discretize the Gaussian distribution, filtering out semantically irrelevant prompts to improve the generation of more meaningful text options, as detailed in Appendix H.
> However, we acknowledge the challenge of projecting sampled embeddings from the prompt distribution into natural language in the current version.
> Despite these challenges, we are committed to enhancing visualization. Thus, we visualize the projection of natural language into embeddings to examine its relationship with the modeled prompt distribution. Specifically, in our visual experiment presented in **Figure R2** of the rebuttal.pdf, we reduce the dimensionality of embeddings for $c_N$ and 10 test prompts and utilize **PCA** for visualization. This illustration showcases how the test prompts' principal components are discretely distributed within the modeled prompt distribution, indicating a flexible probability of selection for adversarial attacks. This underscores the **adaptability** of our modeling in encompassing a broad spectrum of natural language inputs in the semantic space.
>
> We are grateful for your continued support of our work and the invaluable insights you have shared. Your feedback is crucial in refining our efforts, and we truly appreciate your contributions as we endeavor to enhance our work.

---

> > ### Comment · Reviewer_nNzN · 2024-08-11
> > **Thank you for your response**
> >
> > Thank you to the authors for their efforts in addressing my concerns and conducting additional experiments. I will maintain my original rating.

---

### Official Review · Reviewer_JLDf · 2024-07-13

**Soundness:** 2
**Presentation:** 3
**Contribution:** 2
**Rating:** 5
**Confidence:** 4

**Summary:**

- This work is about a method to craft perturbation images to protect users/artists from personalized text to image diffusion methods (specifically DreamBooth and Textual inversion), that generalize better to unseen prompts than previous works.
- The core algorithm is "Prompt-Agnostic Adversarial Perturbation" (PAP). Instead of crafting the perturbation with a fixed condition prompt, PAP model the prompt distribution as a Gaussian dist and sample prompt embedding from it during the perturbation crafting process.
- The prompt distribution is modeled as a Gaussian using Laplace approximation. The mean is estimated by minimizing the diffusion loss starting from a reference prompt. The variance is approximated using a simplified formula based on the difference between the reference and estimated prompts, and their respective loss values.
- Experiments show PAP outperforms previous prompt-specific methods on metrics like FID, CLIP similarity, and LPIPS across different datasets (CelebA-HQ, VGGFace2, Wikiart) and generalization to unseen test prompts.

(after rebuttal)
The authors addressed my main concerns by including results from newer methods with PAP, and clarifying the inconsistent implementation. Despite some issues regarding efficiency and application to more recent personalized techniques/model, I've increased my score from 4 to 5.

**Strengths:**

- The paper is generally well-written, with consistent improvement on unseen prompts when compare with previous works.
- PAP is simple and easy to use, not very costly to add on top of other protection method
- The explanation of the method is easy to understand
- The paper also includes performance under DiffPure, and the result under DiffPure seems to be positive
- The supplementary is informative, with extensive evaluation

**Weaknesses:**

The experimental evaluation need to be improved:
- Apply PAP to other methods like AdvDM + PAP, not just Anti-DreamBooth ASPL, and other recent method such as Diff-Protect [1] to better demonstrate that PAP is a good plug in to the current methods.
- Lacking a simple baseline of adding Gaussian noise to $c_N$ to show the value of approximating H.
- Questionable metric choices, as using LAION aesthetic score for Celeb-HQ and VGGFace2 datasets is not well-justified for face images. And images generated from protected model often contain high-frequency and colorful patterns, which may lead to unreliable LAION aesthetic scores.
- Also evaluation need to include identity score (as used in Anti-DreamBooth) for face datasets, which is crucial to ensure generated images don't contain user's identity.
- A suggestion would be to add an assessment against recent encoder-based personalized techniques, such as InstantID [2], for a more comprehensive comparison.

**Questions:**

- I wonder if PAP can help reduce the sensitivity to the initial prompt when crafting the perturbation between different domain. For example, when testing on human faces or artwork datasets like Wikiart, how would the performance change if in both dataset, $c_0$ is initialized with a very general term, such as an empty string ""
- Pseudo code seems to be not consistent with the implementation. As in your provided implementation, $c_N$ will be recalculated every step in M, while the pseudo code $c_N$ is calculated only once at the start. I wonder if it could affect the performance of the method

- Nitpick: L296 Typo BROSQUE. Table 4 need to have metrics such as FID, LPIPS to be more comprehensive

**Limitations:**

Since PAP still build upon anti-dreambooth, 4-6 minutues A800 when crafting a set of images is still very impractical. Again, I want to see performance of PAP on top of more recent/efficient methods

[1] Xue, Haotian, et al. "Toward effective protection against diffusion-based mimicry through score distillation." The Twelfth International Conference on Learning Representations. 2023.

[2] Wang, Qixun, et al. "Instantid: Zero-shot identity-preserving generation in seconds." arXiv preprint arXiv:2401.07519 (2024).

---

> ### Author Rebuttal · Authors · 2024-08-06
>
> Dear Reviewer JLDf,
>
> We are grateful for your constructive criticism and insightful comments on our work. We greatly value your feedback and have made significant improvements based on your suggestions:
>
> Q1: Apply PAP to other methods including AdvDM + PAP/Diff-Protect
>
> > Per your advice, we have conducted further experiments by integrating PAP with AdvDM and Diff-Protect methods, in addition to Anti-DB. Additionally, based on your feedback on metrics, we have also included additional identity metrics such as the detectable face rate (FDFR) and identity score (ISM) in Anti-DB. The consistent performance showcased in **Table R3**, surpassing baselines across metrics, validates PAP as an effective general plug to enhance the robustness of adversarial attack methods with minimal computational cost.
>
> > **Table R3**: Integrating PAP with AdvDM/Diff-Protect on the VGGFace2 dataset.
> | Method| LPIPS($\uparrow$) | FDFR($\uparrow$) | ISM($\downarrow$) | BRISQUE($\uparrow$) | Time($\downarrow$) |VRAM($\downarrow$) |
> |---|---|---|---|---|---|---|
> |AdvDM|0.66|0.58|0.42|31.41|262s|22G|
> |/.+PAP|0.69|0.61|0.40|33.43|270s|22G|
> |Anti-DB|0.69|0.65|0.38|28.95|288s|28G|
> |/.+PAP|**0.70**|**0.67**|**0.34**|35.02|297s| 29G|
> |Diff-Protect|0.66 |0.63|0.44|31.72|191s|16G|
> |/.+PAP|0.69|**0.67**|0.37 |**36.70**|198s|17G|
> |No Defense|0.55|0.05|0.52|25.67|-|-|
>
> Q2: Lacking a simple baseline of adding Gaussian noise to $c_N$ to show the value of approximating H.
> > We have conducted a simple baseline experiment by directly adding Gaussian noise (with variances 1, 5, 10, and 20) to $c_N$ to evaluate the value of approximating H. As shown in the **Table R4**, our proposed PAP method, with variance estimate $\sigma=H$, achieves the best performance across all metrics. Specifically, it outperforms the second-best method by **3\%($\uparrow$), 2\%($\uparrow$), 3\%($\downarrow$), 2.27($\uparrow$)** on LPIPS, FDFR, ISM, BRISQUE metrics.
> These findings underscore the necessity of the estimation of variance $H$, to generate more effective adversarial perturbations.
>
> > **Table R4**: Simple Baseline of Adding Gaussian Noise to $c_N$ on VGGFace2 datasets
> Variance | LPIPS($\uparrow$)|FDFR($\uparrow$)|ISM($\downarrow$)|BRISQUE($\uparrow$)
> ---|--- |---| ---|---
> 1 |0.67|0.64|0.38|31.92
> 5 | 0.67 | 0.65 | 0.38 |32.75
> 10 | 0.66 | 0.62 |0.40 |29.21
> 20 | 0.64 | 0.60 |0.44|27.01
> $H$ | **0.70**| **0.67**|**0.34** | **35.02**
>
> Q3: Evaluation need to include identity score for face datasets
> > In **Table R3**, we have added the calculation of the detectable face rate (FDFR) and identity score (ISM). In both of these metrics, method + PAP continues to significantly outperform the method itself, including the recent method Diff-Protect.
>
> Q4: Sensitivity to the initial prompt
> > In **Table R5**, we present the outcomes when $c_0=$"", showcasing a significant decline in performance compared to the original PAP. This is attributed to:
> a) $c_0$ serves as a crucial initialization prior for estimating $c_N$, facilitating rapid iteration (only 20 steps) to achieve a reliable approximation of $c_N$;
> b) $c_0$ is involved in the modeling of H estimation. The approximate expression for H estimation is based on the Taylor expansion modeling of $c_0$ and $c_N$.
>
> > **Table R5**: Results with $c_0=$""
> Dataset | FID($\uparrow$)|CLIP-I($\downarrow$)|LPIPS($\uparrow$)|LAION($\downarrow$)|BRISQUE($\uparrow$)|CLIP($\downarrow$)
> ---|---|---|---|---|---|---|
> Celeb-HQ|154.57| 0.72|0.50|5.83|30.18|0.32
> VGGFace2|232.34|0.61|0.60|5.65|29.20|0.29
> Wikiart |320.79|0.69|0.71|5.72|31.88|0.30
>
> Q5: Inconsistency between pseudo code and implementation
> > Initially, we recalibrated the prompt distribution model for each iteration, as reflected in the supplementary material code. Subsequently, we optimized this process and derived the algorithm in the paper, with all experimental results based on this improved algorithm. We intend to upload the code for the current version soon.
>
> Q6: Table 4 in the paper need to have metrics such as FID, LPIPS to be more comprehensive
> > Per your advice, we have supplemented the evaluation with results for FID, CLIP-I, and LPIPS metrics to provide a more comprehensive assessment, as shown in **Table R6**.
>
> > **Table R6**: Performance comparison after applying DiffPure
> |       | FID(↑)| CLIP-I(↓)| LPIPS(↑)|
> |---|---|---|---|
> | AdvDM+DiffPure|301.22(91.48-)| 0.71(0.06+)| 0.70(**0.06-**)  |
> | Anti-DB+DiffPure|335.94(50.46-)| 0.69(0.04+)| 0.68(**0.06-**)  |
> | IAdvDM+DiffPure|271.02(**118.98-**)| 0.72(0.01+)| 0.68(0.03-)|
> | PAP+DiffPure|**379.60**(68.70-)| **0.64(0.08+)**| **0.72(0.06-)**|
> | No Defense|198.71| 0.77|0.62|
>
> Q7: 4-6 minutues A800 when crafting a set of images is impractical.
> > In **Table R3**, we present the time and VRAM required for each method, including the time required when integrating PAP with other methods. The results demonstrate that PAP introduces an average computation time of processing a set of images (<300s) and memory consumption (<30G). In the future, we aim to further optimize the algorithm to reduce the time to within 4 minutes and lower memory consumption to below 24G, thus enabling the model to be used on the GTX 3090.
>
> Q8: Add an assessment against recent encoder-based personalized techniques, such as InstantID.
> > In Table 3 of our paper, we have demonstrated the results on popular customized models such as LORA, TextInversion, and Dreambooth, effectively illustrating the generalizability of our method across different customized models. However, to further continuously validate the reliability and generalizability of our method, we will include results for the InstantX series (including InstantID, InstantStyle) in our future work.
>
> We have made an effort to address your questions and have added additional experiments based on your guidance. We look forward to engaging in fruitful discussions with you and welcome any suggestions for improvement. I hope you will consider providing a more favorable evaluation. Thank you.

---

> ### Comment · Reviewer_JLDf · 2024-08-10
> **Addressed my concerns**
>
> I appreciate the additional experiments and clarifications you provided. In general, the performance when applying PAP to recent methods seem to be positive, and the authors also provide the missing metrics. I have few comments:
> - Regarding Gaussian noise baseline, I notice the variance magnitude tested (1, 5, 10, 20) are quite high with a trend that the higher variance the worst. I wonder if the variance is smaller (0.1, 0.5) could yield improve performance.
> - I expect that PAP could help in term of initial prompt sensitivity when crafting protection on different domains (since the first step is approximating $c_n$). But the quality just slightly lower than the Clean version make me wonder that incorporate PAP could make the protection even more sensitive to the initial prompt (usually a string like "a photo of a person" for human face domain - which we both know that it is not optimal choice)
> - One additional question: For Stable Diffusion 2.x models, which incorporate masking in the text encoder (unlike the fixed 77 tokens cross attention in SD1.x), how does the PAP algorithm adapt to handle variable-length prompts?
>
> (Additional comments)
> - In my opinion, current protection methods still struggle against recent encoder-based personalization, since these personalized methods enable Unet to condition not only on the input latent (which usually OOD because of the added adversarial noise), but also the additional visual information from other encoder(s), so it can by pass the protection. To effectively protect against this threat, incorporating these encoder(s) into the protection mechanism could be a potential future work that you might try.
> - The community is moving towards more recent models, methods, and it's like a cat and mouse game. While PAP shows promising in improving protection on SD, I encourage the authors to continue working on and improving the method, particularly as new personalization techniques and new models emerge
>
> While I still have some concerns, the authors did clarify most of my questions and provide results for the missing experiments. I will raise my score to 5

---

> > ### Author Response · Authors · 2024-08-11
> > **Thanks for your feedback. We hope we can further address your concerns and engage in constructive discussions.**
> >
> > Thank you for your valuable reply. We are glad to address the additional questions you've raised, and we hope our responses will further address your concerns and facilitate constructive discussions on the future work of PAP.
> >
> > > Q1: If the variance is smaller (0.1, 0.5) could yield improved performance.
> >
> > Firstly, within the tested variance range (1, 5, 10, 20), the performance trend is indeed a slight initial rise (from 1 to 5) followed by a decline (from 5 to 20). Additionally, to better address your concern, we have supplemented **Table R7** with control experiments using variances of 0.1 and 0.5 as you mentioned.
> >
> > **Table R7**: Simple Baseline of Adding Gaussian Noise
> > | Variance | LPIPS(↑) | FDFR(↑) | ISM(↓) | BRISQUE(↑) |
> > |----|-----|------|---------|------|
> > | 0.1        | 0.66     | 0.62    | 0.41 | 30.98|
> > | 0.5       | 0.66     | 0.63    | 0.39    | 31.74|
> > | H        | 0.70     | 0.67    | 0.34 | 35.02|
> >
> > > Q2: Incorporate PAP could make the protection even more sensitive to the initial prompt.
> >
> > Regarding the quality in **Table R5** being slightly lower than the Clean version, we note that these results used the same setup as described in the paper: 20 gradient descent steps starting from "" to obtain $c_N$. We plan to adapt our approach by dynamically tuning the learning rate and stepsize. Moreover, instead of using an empty string, further experiments are needed to identify a generic $c_0$ and optimization settings, thereby eliminating the need to manually select prompts for different tasks.
> >
> > > Q3: How does the PAP algorithm adapt to handle variable-length prompts?
> >
> > Currently, for SD2.x, our approach is to pad all prompts to a fixed length of 77 tokens for processing, consistent with our approach for SD1.x. Although this may not be the optimal solution, we adopted this strategy for the following reasons:
> > 1. Prompt Length Sufficiency: In general, a length of 77 tokens is adequate to accommodate most prompts, as they rarely exceed this limit in practical applications.
> > 2. Optimization Landscape Smoothness: Maintaining a consistent prompt length ensures a smoother optimization landscape, which is advantageous for solving the optimization problem effectively.
> >
> > Despite this approach's potential limitations, the results in **Figure R1** from the rebuttal.pdf demonstrate that our method still achieves satisfactory performance.
> >
> > > Additional comments1: Incorporating more encoders into the protection mechanism
> >
> > While the paper considers attacks using the VAE encoder, which is one of the most popular and widely used image encoders in SD, other encoders are worth considering for integration into the attack process. This could help enhance the generalizability of the proposed method.
> >
> > To this end, in our future work, we will further design attack strategies for image encoder-agnostic scenarios. Specifically, we will consider several of the most commonly used image encoders, and by integrating these encoders with our modeling of image feature distributions, we will sample and attack the feature distributions of images across various encoders. This will be further
> >  fused with our original PAP approach in the future.
> >
> > We anticipate that the above proposed **Prompt & Encoder-Agnostic Perturbation (PEAP)** method will be robust against various attack prompts and image encoders, aligning with our pursuit of image protection that better reflects real-world scenarios.
> >
> > > Additional comments2: Continue working on and improving the method, as new personalization techniques and new models emerge
> >
> > Thank you for your encouragement. We are aware of the rapid development in the field of customized generation, such as the InstantX series. We will continue to enhance PAP by experimenting with new personalization techniques and models, while also striving to improve the attack efficiency and reduce the computational cost of PAP.
> >
> > Finally, we sincerely appreciate your suggestions and encouragement for our work. We welcome further discussions on the ideas mentioned, and we are deeply grateful for your willingness to increase your score.

---

> > ### Author Response · Authors · 2024-08-13
> > **A gentle reminder**
> >
> > Dear Reviewer JLDf,
> >
> > We greatly appreciate your comprehensive review and valuable feedback.
> > As noted in your recent response, we have addressed most of your concerns, and you kindly expressed your intention to raise the score.
> >
> > With the discussion deadline approaches, we respectfully remind you of your previous commitment to increase the score and sincerely request your final assessment.
> >
> > Best,
> >
> > Authors

---

### Official Review · Reviewer_JA4c · 2024-07-17

**Soundness:** 3
**Presentation:** 2
**Contribution:** 2
**Rating:** 6
**Confidence:** 4

**Summary:**

The authors propose a prompt-agnostic adversarial perturbation (PAP) method for customized diffusion models. They first use Laplace approximation to model the prompt distribution. Then they derive the attacks by maximizing the disturbance expectation. Extensive experiments on three datasets validate their performance.

**Strengths:**

Experiments on three datasets.

Interesting topic.

Mathematical Proof.

**Weaknesses:**

Potential semantic gap between the estimated prompts and natural language.

Experiment Scope: only do experiments on SD 1-5.

**Questions:**

1 This paper only mentions the time complexity of their approach in Line 226, but what are the time comparisons with baselines?

2 Only considering SD 1-5, however more versions of stable diffusion models and other diffusion models should also be considered. Could you please provide more results on other models to show the generalization of your approach?

3 The authors only show the average performance across 10 similar prompts. What is the performance of the trained prompts?

4 The authors use a Laplace to approximate the prompt distribution, but why is it a proper approximator? Additionally, in my point of view, the approximation is in the semantic space. However, the users will input natural language. Therefore, how to combat the domain gap is another concern.

**Limitations:**

Yes

---

> ### Author Rebuttal · Authors · 2024-08-06
>
> Dear Reviewer JA4c,
>
> Thank you for your valuable feedback and insightful comments on our work. We appreciate your suggestions and have made the following improvements:
>
> Q1: Potential semantic gap between the estimated prompts and natural language
> > a) We emphasize the importance of considering prompt samples that are not only close to the mean but also embody meaningful semantic information to bridge the semantic gap between the estimated prompt distribution and natural language in Appendix H.1. As part of our future work, a proposed restriction module could discretize the Gaussian distribution, thereby sieving out prompts devoid of semantic relevance to facilitate the generation of more meaningful text options, as detailed in Appendix H.
>
> > b) To further explore the relationship between the estimated prompts and natural language, we have conducted a visual experiment in **Figure R2** of the rebuttal.pdf. By reducing the dimensionality of 10 test prompts' embeddings and $c_N$ using PCA, we visualize the estimated prompt distribution and test prompts projected in a low-dimensional space for the tasks of "Facial Protection" (left) and "Preservation of Artistic Style" (right) respectively. The figures demonstrate that the two principal components of test prompts are discretely distributed within the modeled prompt distribution, indicating a flexible probability of being selected for adversarial attacks. This illustrates that our modeling effectively covers a range of natural language inputs in the semantic space.
>
> Q2: More versions of stable diffusion models should be considered.
> > Per your advice, we have conducted additional experimental evaluations on the Wikiart dataset using SD1.4 and SD2.0.
> The results in **Figure R1** of the rebuttal.pdf indicate that the PAP method continues to show significant improvements on various diffusion versions, demonstrating the generalization of our approach.
>
> Q3: Time comparisons with baselines
> > In **Table R1**, we present the time required for each method. The results demonstrate that PAP introduces an average computation time of processing a set of images (<300s). In the future, we aim to further optimize the algorithm to reduce the time to within 4 minutes.
>
> > **Table R1**: Time and VRAM comparisons with baselines.
> | Method | AdvDM | Anti-DB |IAdvDM | PAP|
> |---|---|---|---|---|
> | Time| 262s|288s|204s|297s
>
> Q4: What is the performance of the trained prompts?
> > In Table 6 and Figure 5 of Appendix F, we have already presented the performance of previous methods for each prompt, including the trained prompts. Furthermore, we have extended the display of the performance of the trained prompts for all methods. In **Table R2**, our method slightly trails behind the SOTA by 0.01/39.57 in LPIPS/FID respectively. However, our method still maintains a leading position in ISM, FDFR, BRISQUE, and CLIP metrics.
>
> > **Table R2**: Performance of the trained prompts.
> | Method| LPIPS($\uparrow$) | FDFR($\uparrow$) | ISM($\downarrow$) | BRISQUE($\uparrow$) | FID($\uparrow$) | CLIP($\downarrow$) |
> |---|---|---|---|---|---|---|
> | AdvDM| 0.65| 0.61| 0.39| 33.68| **301.55**| 0.25|
> | Anti-DB| **0.71**| **0.68**| 0.34| 32.24| 277.24| **0.24**|
> | IAdvDM| 0.65 | 0.57| 0.43| 33.55| 296.31| 0.28|
> | PAP| 0.70|**0.68**| **0.33**| **36.43**| 261.98| **0.24**|
> | No Defense|0.50| 0.01| 0.55| 23.22| 128.31| 0.38|
>
> Q5: Why is  Laplace Approximation a proper estimator?
> > a) Approximation of Gaussian distributions: The Laplace approximation often yields a Gaussian distribution. In many cases, especially for large amounts of observed data and problems applicable to the central limit theorem (which aligns with the prompt embedding space), the true distribution may approximate a Gaussian distribution. This implies that, for large samples, the Laplace approximation provides a good asymptotic approximation of the shape of the probability density function;
>
> > b) Computational simplification:  Compared to more complex methods such as Monte Carlo simulations, the Laplace approximation often has computational advantages. It provides a relatively straightforward way to approximate the true distribution, especially when the analytic form of the distribution is difficult to handle or unavailable (our ideal prompt distribution is challenging to solve analytically as discussed in Section 3.2). This simplification makes the Laplace approximation practical in real-world problems.
>
> > c) We analyze the properties that the ideal prompt embedding distribution should meet: **P1**. The distribution should be centered around the extreme points $c_x$, with probability decreasing as semantic relevance diminishes, and with a large number of samples; **P2**. The analytical form of the distribution is unavailable. **P1** aligns with the use of Gaussian distribution for approximation, as discussed in a), while **P2** corresponds well with the situation described in b). These indicate that the scenario we are addressing is highly suitable for Laplace modeling.
> Subsequently, by Taylor expanding at $c_x$, we determine an approximate Gaussian distribution (Section 3.3.1) and introduce two estimators (Section 3.3.2): one minimizing the generation loss and the other performing a Taylor expansion around $c_x to estimate their mean and variance, respectively. We also provide a detailed explanation of the error bounds resulting from the approximation of these estimators in Appendix A, thereby theoretically supporting the validity of our Laplace estimation. Furthermore, the extensive experimental results presented in the paper (Tables 1/3/4/10, Figures 2/3/4/6, visualization results in Appendix I) empirically reinforce the validity of our approach.
>
> In conclusion, we look forward to further discussions and insights on our work. Your feedback has been invaluable in shaping our research, and we are eager to continue this dialogue. Thank you for your time and consideration.

---

> > ### Comment · Reviewer_JA4c · 2024-08-14
> >
> > Thanks for the rebuttal and parts of my concerns are addressed. I raise my score to 6.

---

### Author Rebuttal · Authors · 2024-08-06

Dear AC and all the reviewers,

We would like to express our sincere gratitude to you for your comprehensive evaluation of our manuscript, as well as your insightful feedback and constructive suggestions.

We have tried our best to answer all questions of the reviewers about our paper. We wander if our responses address all the concerns?

Thanks all !

---

### Decision · Program_Chairs · 2024-09-25

**Decision:**

Accept (poster)

**Comment:**

This paper was reviewed by four experts in the field. The initial scores were Borderline Accept, Borderline Reject, Accept, and Borderline Accept. The rebuttal addressed part of the reviewers' concerns and convinced 2 reviewers to increase their scores, leading to all acceptance recommendations. The reviewers applauded the paper for its well-motivated problem, good writing, simple and easy-to-use technique, extensive and convincing experiments. Based on the reviewers' feedback, the decision is to recommend the paper for acceptance to NeurIPS 2024. The reviewers did raise some valuable concerns that should be addressed in the final camera-ready version of the paper. The authors are encouraged to make the necessary changes to the best of their ability. We congratulate the authors on the acceptance of their paper!